# Phased-assembly-driven pangenome graphs for structural variant genotyping and complex trait mapping in dairy cattle

Liu Yang [1,2], Yahui Gao[1,2], Kristen L. Kuhn [3], Nayan Bhowmik[1], Wenli Li[4], Geoffrey Zanton [4], Lingzhao Fang [5], John B. Cole [6,7,8,9], Congjun Li[1], Ransom L. Baldwin, VI [1], Curtis P. Van Tassell [1], Benjamin D. Rosen [1], Li Ma [2], Timothy P. L. Smith [4] ✉ & George E. Liu [1] ✉

Structural variants are an underexplored source of genetic diversity. As part of the FarmGTEx Project, here we report a Holstein breed-specific pangenome graph (H20D) using Minigraph-Cactus and 40 phased haploid assemblies from 20 cows. H20D outperforms both assembly- and read-based long-read callers, and far exceeds short-read approaches, identifying over 10,000 additional structural variants per sample. It also significantly improves structural variant detection and genotyping relative to graphs built across breeds or from fewer/unphased assemblies, with particular advantages in complex regions. Using H20D, we genotype variants in 173 cattle and performed a GWAS, where a larger fraction of structural variants than SNPs reach genome-wide significance, implicating them as potential causal variants. Together, these results demonstrate the power of phased, within-breed pangenome graphs for accurate SV genotyping and trait mapping in dairy cattle.

Understanding the full spectrum of genetic diversity is crucial for improving human health[1–3] and livestock traits[4–6]. Although traditional genome-wide association studies (GWAS) have primarily focused on single-nucleotide polymorphisms (SNPs), structural variants (SVs), including insertions, deletions, inversions, and more complex rearrangements, represent a significant and underexplored component of genetic variation[7–19].

While single-reference genomes serve as essential tools for genetic analysis, they are inherently limited in capturing the full range of genetic diversity within a species, especially for SVs, which vary significantly across populations[20]. This limitation has driven the development of pangenomes with graph-based representations that incorporate multiple genomes to better capture sequence diversity.

The human pangenome, built in 2010 using two non-European genomes[21], identified ~40 Mb of additional sequences, with subsequent studies revealing an additional 30–50 Mb not present in the short-read-based human reference[22,23]. Long-read-based human assemblies uncovered over 850 Mb of structurally variant DNA[24,25]. The Genome in a Bottle (GIAB) project later introduced regional stratifications for human references, distinguishing "difficult" regions, those challenging to map due to repetitive sequences or GC-rich content, from confident regions[26–28].

Short-read sequencing often produces fragmented assemblies, especially in repetitive regions containing segmental duplications or SVs. To address this, the Telomere-to-Telomere (T2T) Consortium generated gapless human assemblies using long-read technologies like

¹Animal Genomics and Improvement Laboratory, Beltsville Agricultural Research Center, Agricultural Research Service, United States Department of Agriculture, Beltsville, MD, USA. ²Department of Animal and Avian Sciences, University of Maryland, College Park, MD, USA. ³USDA, ARS, U.S. Meat Animal Research Center (USMARC), Clay Center, NE, USA. ⁴US Dairy Forage Research Center, USDA-ARS, Madison, WI, USA. ⁵Quantitative Genetics and Genomics (QGG), Aarhus University, Aarhus, Denmark. ⁶Council on Dairy Cattle Breeding, 4201 Northview Dr, Bowie, MD, USA. ⁷Department of Animal Sciences, Donald Henry Barron Reproductive and Perinatal Biology Research Program, and the Genetics Institute, University of Florida, Gainesville, FL, USA. ⁸Department of Animal Science, North Carolina State University, Raleigh, NC, USA. ⁹Centre for Genetic Improvement of Livestock, Department of Animal Biosciences, University of Guelph, Guelph ON, Canada. ✉e-mail: tplsmith@gmail.com; George.Liu@usda.gov

PacBio and Oxford Nanopore Technologies (ONT), complemented by Hi-C[20,29–31]. These efforts, along with the Human Pangenome Reference Consortium (HPRC), which seeks to sequence and assemble 350 diverse genomes[32–34], highlight the limitations of single linear references and the value of pangenome graphs for comprehensive variation detection[20,35]. Tools such as vg[36], Minigraph[37], MC (Minigraph-Cactus)[38], and the PGGB (PanGenome Graph Builder)[39], have been developed to construct pangenome graph assemblies, each with specific advantages and trade-offs[40,41].

Variation-aware graphs are highly useful for SV calling, with nodes representing shared core sequences and bubbles depicting alternative variations, including complex SVs among individuals. A path through the graph can be transformed into a haplotype sequence. Tools like PanGenie[42] and Giraffe (from vg)[43] were designed to genotype genetic variants, including SVs, using short reads and long-read-based SV catalogs[44]. A recent graph-based human SV study sequenced 1019 long-read genomes from 26 populations in the 1000 Genomes Project, identifying 167,291 sequence-resolved SVs and revealing mechanisms like LINE-1 and SVA transductions[45]. A companion paper reported a multi-ancestry SV imputation panel from 888 of the 1019 samples[46]. These findings provide insights into SV formation, especially involving repeat sequences and homology-mediated rearrangements, demonstrating the impact of long-read sequencing on understanding genomic architecture and disease.

SVs have been shown to influence a range of complex traits in cattle, including body weight, milk yield, reproductive fitness, and disease resilience, making them important targets for breeding programs[47–50]. In cattle, pangenomic approaches provide a more complete view of SV diversity, facilitating improved SV detection and crossbreed comparisons[51–57]. For example, variation-aware genome graphs have been shown to substantially improve variant detection and genotyping relative to linear references[51,52], while breed-specific and multiassembly graphs enhance read mapping, uncover additional functional sequences, and capture previously hidden structural variant[53,54]. Large-scale graph genomes incorporating global breed diversity further expand the resource for studying genetic variation across populations[55,56]. Leonard et al. (2024) demonstrate that genotyping SV using a cattle pangenome enhances the resolution and accuracy of molecular phenotype mapping[57]. The Holstein breed, known for its high productivity and rich phenotypic records, presents an ideal model for pangenomic analysis[58,59]. We have performed long-read and preliminary pangenome analyses using Minigraph[37], revealing within-breed SVs and additional sequences in Holstein and Jersey cattle[47]. Collectively, these studies highlight the value of graph-based approaches for accurate SV representation, functional interpretation, and precision breeding in cattle.

Despite these advances, current cattle pangenomes still have limitations. Sensitivity to haplotype resolution and the representation of complex SVs remains low. Many graphs focus on single-reference augmentation rather than fully phased SV variation, and most are constructed from a limited number of assemblies per breed, restricting capture of the full SV spectrum. Consequently, SV coverage remains incomplete, limiting high-resolution genotyping and trait association studies, motivating the need for a comprehensive, haplotype-resolved within-breed pangenome.

In this study, we used the Minigraph-Cactus pipeline[38] with 40 phased haploid assemblies from 20 Holstein cows to construct a within-breed pangenome graph (H20D) and benchmark it against multiple other approaches. H20D not only outperformed short-read methods but also provided a more comprehensive SV profile than assembly- or read-based long-read callers. Compared to cross-breed graphs or those built from varying assembly counts and types (unphased vs. phased), H20D markedly improved SV detection and genotyping. We further performed GWAS using both SNP and SV genotypes, demonstrating the contribution of SVs to traits such as production and body weight. Overall, our findings highlight the value of optimized within-breed pangenomes for accurate and efficient SV genotyping in large whole-genome sequencing cohorts.

## Results

### Single-breed pangenome graph for Holstein (H20D)
We constructed the Holstein-specific pangenome graph (H20D) using the Minigraph-Cactus (MC) pipeline, with a phased haplotype assembly model and the cattle reference genome ARS_UCD2.0 as the backbone. This pangenome graph was built from 40 phased haploid assemblies (average genome size of 3.11 Gb and average contig N50 of 30.12 Mb) derived from 20 Holstein cows (average read length ranging from 13 to 23 kb and coverage between 22.5× and 32.6×), all sequenced using PacBio HiFi reads (Supplementary Fig. 1, Supplementary Data 1 and Supplementary Data 2). Two of these samples were also subjected to ~40× coverage Hi-C sequencing. One sample (Sample 4611) with Hi-C achieved contig N50 values of 63.48 Mb and 52.52 Mb for haplotypes hap1 and hap2, respectively. On average, Hi-C sequencing increased contig N50 by 5.67 Mb for the 4 phased haplotype assemblies.

The autosomal size of H20D, including the reference genome ARS_UCD2.0, was 2.56 Gb, with a total genome length of 2.86 Gb (Fig. 1a and Supplementary Data 3). After excluding 142.84 Mb ARS_UCD2.0 unique sequences, H20D contained 51.01 million (M) non-reference nodes and 69.14 M edges in its 2.71 Gb, with a calculated graph complexity of 1.364 (the ratio of edges to nodes) for autosomes (Supplementary Fig. 2 and Supplementary Data 3-4). But its unplaced chromosomes (chrOther) showed a higher complexity of 1.402. When calculating pangenome sizes based on individual Holstein genomes, we found that 94.9% (2.58 Gb) of the sequences (i.e., core) were shared across all 20 Holstein samples, while 5.1% (138.68 Mb) represented variable sequences (Fig. 1b and Supplementary Fig. 2). Among these variable sequences, an average of 1.23 Mb per individual, totaling 24.70 Mb (0.91%), was unique to each Holstein sample (singletons). At the haplotype level, only 61.1% (1.66 Gb) of the genome was shared across the 40 assemblies, with 38.9% (1.06 Gb) representing variable regions, which included 23.69 Mb (0.87%) of sequences unique to each assembly, with an average size of 0.6 Mb per assembly (Supplementary Fig. 2).

### Genomic variations in H20D pangenome graph relative to ARS_UCD2.0
We then performed a classification of genomic variations in H20D relative to the ARS_UCD2.0. The identified variation types included single-nucleotide variants (SNVs), insertions (INS), deletions (DEL), and complex rearrangements (COMPLEX). "COMPLEX" refers to variant alleles located within pangenome-graph bubbles containing more than two branches, representing sequence changes of at least two base pairs that cannot be classified simply as DEL or INS[60]. Variants with a genotyping rate above 10% were kept. After filtering, from autosomes, we obtained 47.76 M nodes and 65.16 M edges, covering 12.77 M SNVs, 1.55 M DEL, 1.60 M INS, and 0.61 M complex (COMPLEX) events (Fig. 1d and Supplementary Data 5). To account for nested and structurally complex variations, we used PanGenie's graph-based genotyping framework to decompose multi-allelic sites into standardized bi-allelic representations[60]. On average, this conversion led to a 5.51% increase in alleles, with the most significant increase (7.71%) observed in COMPLEX variants. (Supplementary Data 5). The total autosomal lengths of DEL, INS, and COMPLEX events (summing each allele) were 35.52, 115.75, and 132.10 Mb, respectively, corresponding to 89.00 Mb of non-redundant sequences, accounting for 3.58% of the cattle reference genome (Fig. 1e, f and Supplementary Data 5). On a per-sample basis, we detected an average of 6.25 M SNVs, 0.54 M DEL, 0.61 M INS, and 0.22 M COMPLEX variants. The total lengths of DEL, INS, and COMPLEX variants were 14.63 Mb, 16.20 Mb, and 12.32 Mb, respectively, corresponding to 0.25%, 0.59%, 0.65%, and 0.49% of the cattle reference genome ARS-UCD2.0 (Supplementary Data 6). For SV (≥ 50 bp),

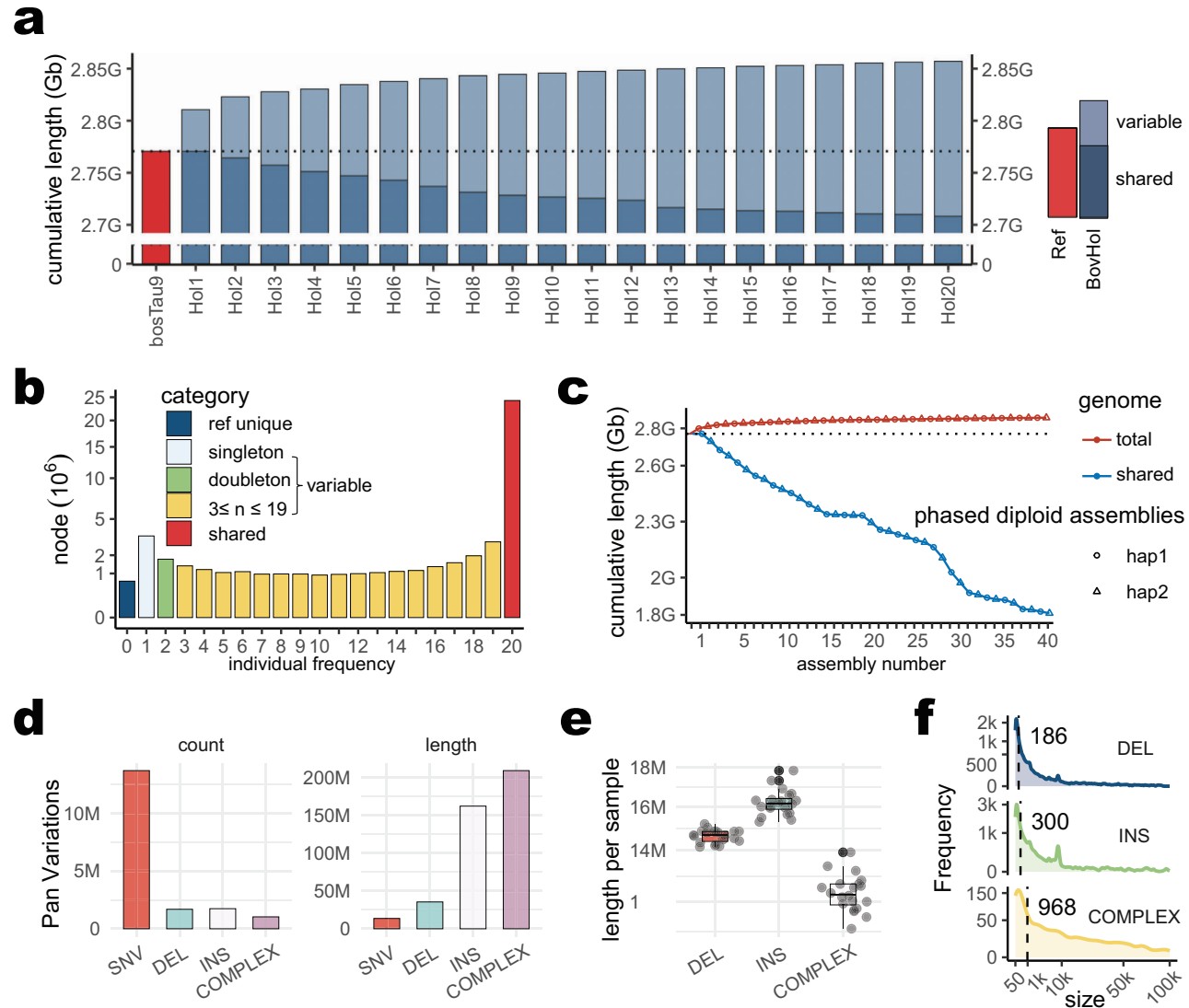

**Fig. 1 | Characteristics of the Holstein cattle pangenome graph H20D.**
**a** Pangenome cumulative length curves for the H20D pangenome graph, sorted into shared and variable sequences by samples. The dotted line indicates the size of the ARS_UCD2.0 reference genome. **b** Classification of pangenome nodes in Holstein cattle. **c** Pangenome cumulative length curves for H20D, stratified by phased diploid assemblies (hap1 and hap2). The dotted line represents the ARS_UCD2.0 reference size. **d** Variation counts and lengths in H20D. **e** Length distributions of three SV types (DEL, INS, and COMPLEX) across 20 Holstein samples. Boxplots show the median (center line) for 20 independent individuals (n = 20), interquartile range (IQR; box), and data within 1.5×IQR (whiskers). All individual sample values are displayed as points, and no outliers were removed. Each point represents the total length of DEL, INS, or COMPLEX variants in one sample. **f** Count distributions of 3 SV types in H20D. Dash lines and numbers show the average size of each type.

we observed an enrichment at chromosome ends (within 5 Mb), where deletions and insertions were nearly twice as frequent as in other genomic regions (Supplementary Fig. 3).

## Comparison of Holstein and crossbreed pangenome graphs (H20D vs. M13H)

To construct the crossbreed pangenome graph (**M13H**), we downloaded and incorporated published high-quality assemblies (average genome size of 2.96 Gb and average contig N50 of 48.64 Mb) from 13 different cattle breeds, including 8 *Bos taurus* breeds (Jersey, Holstein, Charolais, Braunvieh, Brown Swiss, Piedmontese, African N'Dama, and Hanwoo), 2 *Bos indicus* breeds (African Ankole, and Nellore), and 2 mixed-blood breeds (Yunling and Mongolian) (Supplementary Fig. 4a and Supplementary Data 6). The M13H crossbreed pangenome graph, which included one Holstein assembly, was 2.94 Gb, 0.08 Gb larger than H20D, with an autosomal size of 2.62 Gb (Supplementary Fig. 4a and Supplementary Data 7–8). The M13H pangenome exhibited

significantly more non-reference elements, containing 94.39 M nodes and 129.24 M edges, nearly double the node and edge counts of H20D (Supplementary Fig. 4b and Supplementary Data 7–8). The overall pangenome complexity increased slightly from 1.364 in H20D to 1.369 in M13H (Supplementary Fig. 4c). When calculating individual pangenome sizes based on individual genomes, M13H contained 3.80% (111.57 Mb) of unique sequences per individual, while 86.80% (2.55 Gb) was shared across breeds (Supplementary Fig. 4d).

The SV landscape in M13H was broader than in H20D, with 26.28 M SNVs, 3.60 M deletions, 2.93 M insertions, and 1.43 M complex variants (Supplementary Data 9–10). Converting multi-allelic into bi-allelic variants resulted in a smaller proportional increase of 2.93% in M13H, compared to a 5.51% increase in H20D. This effect was most pronounced in COMPLEX variants, where the increase was 7.71% in H20D but only 2.74% in M13H. (Supplementary Data 9). The M13H graph exhibited a higher total length of DEL (57.19 Mb), and a similar total length of INS (118.54 Mb), but notably 41.22% less COMPLEX

variant length (77.65 Mb) compared to H20D (Supplementary Fig. 5a). The total non-redundant SV length in M13H reached 154.82 Mb, 40.51% greater than in H20D, covering 6.22% of the cattle reference genome (Supplementary Fig. 5a).

Since one Holstein primary assembly (ARS − LIC_NZ_Holstein −Friesian_1) was included in M13H, we migrated it from the M13H group (now M12H) into the Holstein group (H20D1H) for variation comparisons. As expected, M12H contained a higher proportion of unique variants (55.30%) compared to H20D1H (20.62%) (Supplementary Fig. 5b and Supplementary Data 10). While counts of specific variants were similarly distributed across variant types, their lengths differed (Supplementary Fig. 5b and Supplementary Data 10). For example, the total H20D1H-specific INS lengths (86.6 Mb) were comparable to that of M12H (95.3 Mb), whereas H20D1H exhibited a greater length of complex variants (90.4 Mb) compared to M12H (63.8 Mb) (Supplementary Fig. 5b and Supplementary Data 10). We observed that larger insertions and complex variants (>1 kb) were more frequent in H20D1H, contributing to nearly equal total unique SV lengths in H20D1H (188.0 Mb) and M12H (196.0 Mb) (Supplementary Fig. 5c, d). It is noted that this section focuses on broad trends rather than definitive biological conclusions, as it is not possible to clearly separate true biological differences from technical variation caused by differences in assembly quality or batch effects.

## Evaluation of long-read SV calling tools versus pangenome strategies

We compared SV calls using a pangenome-based approach (Minigraph-Cactus pipeline for pangenome graph generating and *vg deconstruct* for the VCF exporting, MC) with four assembly-based strategies (SVIM-asm[61] and PAV[42] with both diploid and haploid assemblies): SVIM-asm_Diploid, SVIM-asm_Haploid, PAV_Diploid, and PAV_Haploid and 5 read-based strategies (cuteSV[62], pbsv at https://github.com/PacificBiosciences/pbsv, Sniffles[63], SVIM[64], and SVision[65]) across 20 Holstein HiFi datasets. To highlight the distinctive properties and high-confidence variant representation of the pangenome approach, we used the H20D pangenome variants as a baseline truth set for pairwise comparisons. For this analysis, SVs on ChrX from female samples, ranging from 50 bp to 1 Mb, were retained. Both SVIM-asm and PAV can utilize either a haploid model (which requires a primary assembly) or a diploid model (which requires 2 haplotype assemblies). Consequently, we generated 9 comparisons between MC and these strategies, with MC detecting an average of 25.39k SVs per individual, while other strategies ranged from 16.38k (PAV_Haploid) to 35.26k (SVIM). For the assembly-based strategies, diploid models detected 6.96k more SVs on average than haploid models (Supplementary Data 11).

To assess genomic coordinate consistency, we calculated the proportion of overlapping SV regions (≥90%) between MC and the other methods using BEDtools. Among assembly-based approaches, the diploid model had a ~20% higher overlap with MC than their haploid counterparts, with ~66% of MC SVs shared with both SVIM-asm and PAV in the diploid model (Fig. 2a and Supplementary Data 12). In contrast, read-based methods had a broader range of unique SV regions, with MC-specific SVs accounting for 34.35% to 39.63% of their total calls. SVIM exhibited the highest proportion of unique SVs (54.75%) due to its elevated call count (~35.26k SVs per individual). On average, the MC pangenome method identified 10.27k unique SVs (ranging from 8.71k to 13.71k), or 40.43% (ranging from 34.30% to 53.98%), compared to other long-read SV calling methods.

We further evaluated precision, recall, and genotype concordance using Truvari between MC and each tool. Due to input format requirements for reference and alternative allele sequences, SVIM and SVision could not be assessed and were excluded. For the remaining 7 comparisons, the 2 assembly-based methods with a diploid model of PAV and SVIM-asm exhibited the highest consistency with the MC.

Specifically, PAV_Diploid and SVIM-asm_Diploid achieved the highest genotype concordance (0.94 and 0.93), followed by read-based pbsv (0.91), Sniffles, and cuteSV (both at 0.90) (Fig. 2b and Supplementary Data 12). However, the haploid models of SVIM-asm and PAV struggled to genotype SVs effectively, likely due to their reliance on single primary assemblies, which cannot capture the diploid nature and complex structural variant within genomes. This limitation is particularly problematic when analyzing populations with high heterozygosity or diverse genomic backgrounds. PAV_Diploid and SVIM-asm_Diploid also achieved the highest F1 score (0.81 and 0.79) and recall (0.80 and 0.81), while their haploid counterparts (PAV_Haploid and SVIM-asm_-Haploid) showed lower F1 scores (0.67 and 0.64) with recall rates (0.55 and 0.57), but similar precision (0.84 and 0.72).

Building on these insights, we next explore the advantages of diploid pangenome modeling and its impact on SV detection and genotyping accuracy. To investigate discrepancies between MC and the other SV calling tools, we categorized SVs by size (50–200 bp, 200 bp–1 kb, 1–10 kb, 10–100 kb, and >100 kb). The smallest SVs (50–200 bp) had the lowest shared proportion with MC, while shared rates increased for larger SVs (Fig. 2c). The F1 score declined when SVs reached sizes between 10 kb and 100 kb. In terms of genotype concordance, assembly-based tools maintain stable performance for SVs up to 10 kb, whereas read-based tools exhibited a decline for SVs exceeding 200 bp (Fig. 2d). Little variation was observed in shared proportions across SV types (ALL, DEL, or INS) (Supplementary Fig. 6a). In comparison, the F1 score and genotype concordance of DEL (0.79 and 0.95, excluding haploid models for assembly-based methods) are higher than those of INS (0.75 and 0.88) (Fig. 2e), likely because insertions involve more complex sequence contexts, making alignment and validation more challenging. Deletions, by contrast, are easier to detect as they represent the absence of a sequence. Chromosome-level comparisons revealed that Chr1 (68.33%) and Chr6 (68.95%) had the highest shared SV proportions, while ChrX (45.99%) and Chr13 (49.91%) had the lowest. Interestingly, ChrX showed the lowest shared rate (~35.73%) among haploid models of assembly-based methods compared to MC (Supplementary Fig. 6b). When evaluating SV consistency in repeat regions between MC and the other tools, we observed lower concordance in repeat-rich regions than in non-repetitive regions (Fig. 2f). The average F1 score for SVs in non-repeat regions was 0.37 higher than in repeat regions, with a 0.52 increase in precision. Differences in recall rate and genotype concordance were modest (both at 0.12). Satellite regions exhibited the lowest F1 scores (<0.1), and genotype concordance was notably lower in satellite, simple repeat, and low-complexity regions compared to other genomic regions (Fig. 2f).

Overall, the lowest genomic coordinate consistency was observed for short SVs (<1 kb), with consistency improving for larger variants. Genotype concordance varied more for SVs >10 kb, with INS exhibiting slightly lower genotype concordance than DEL. Among genomic features, satellite regions had the greatest impact on overall consistency with MC. ChrX exhibited the lowest shared SV proportions with MC across the 9 assembly-/mapping-based strategies, particularly among haploid models of assembly-based methods like PAV_Haploid and SVIM-asm_Haploid.

## Improved SV detection and genotyping in phased diploid pangenomes compared to haploid pangenomes

We then built a primary assembly-based pangenome graph (H20H) from the same 20 Holstein samples (Supplementary Data 6). Compared to H20D, the autosomal size of H20H decreased by 8.92 Mb (0.35%) to 2.55 Gb, with corresponding reductions in nodes (down 10.84% to 42.58 M) and edges (down 11.01% to 57.98 M) (Fig. 3a and Supplementary Data 7). Variant counts also dropped, including SNVs (−8.89% to 11.64 M), deletions (−20.75% to 1.23 M), insertions (−16.84% to 1.33 M), and complex variants (−26.02% to 0.45 M), collectively decreasing non-redundant sequences by 10.20 Mb (11.46%) to

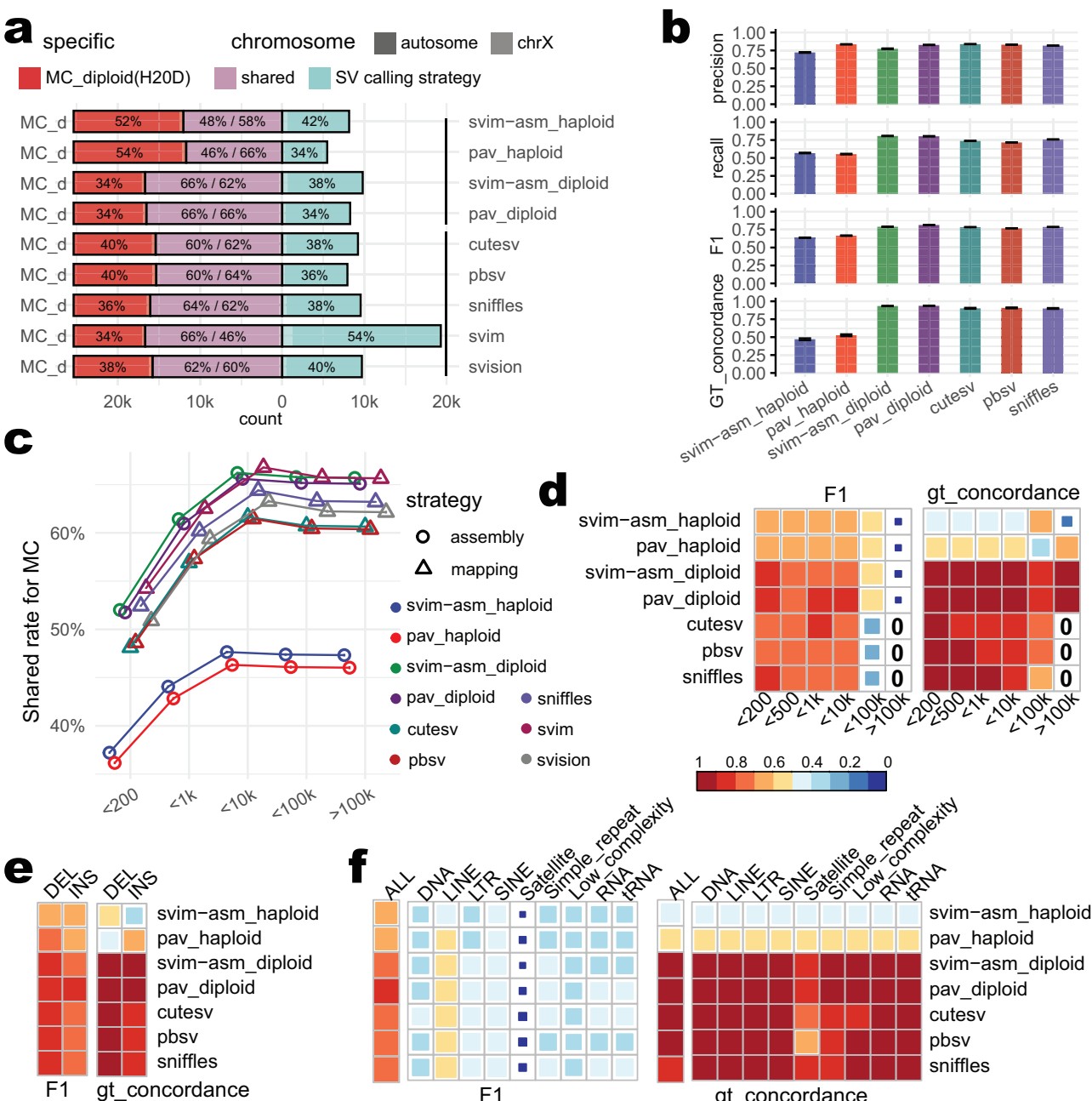

**Fig. 2 | Comparisons of SV calling strategies for 20 Holstein HiFi datasets.** **a** Comparison of SV calls: The H20D pangenome-based strategy serves as the baseline for evaluating 4 assembly-based strategies (SVIM-asm_Diploid, SVIM-asm_Haploid, PAV_Diploid, and PAV_Haploid) and 5 read-based strategies (cuteSV, pbsv, Sniffles, SVIM, and SVision). **b** Performance metrics: Precision, recall, and genotype concordance of other strategies were evaluated against the H20D pangenome-based strategy. Bars indicate mean ± SE (standard error) with n = 20. **c** Proportions of shared SV lengths: The proportions of shared SV lengths between other strategies and the H20D pangenome-based strategy were evaluated for different SV lengths. F1 values and genotype concordances of other strategies were evaluated against the H20D pangenome-based strategies, grouped by different SV lengths (**d**). different SV types (**e**); and different repeat classes (**f**). SVIM and SVision results were omitted in (**b**, **d**–**f**) due to Truvari's requirement for genomic sequence input.

78.81 Mb (Supplementary Data 13). An average of 16.24 k SVs was identified from H20H autosomes versus 24.63 k in H20D, a 34.04% reduction, with lower genotype concordance (F1 = 0.73, precision = 0.91, recall = 0.60) (Fig. 3b and Supplementary Data 14 and 15). Of H20H SVs, 80.74% (13.11 k) overlapped with H20D, while 19.26% (3.13 k) did not. Conversely, H20D captured more unique SVs (10.59 k/ 43.01%), with a shared rate of 56.99% (14.03 k SVs) relative to H20H (Fig. 3c). We evaluated the consistency of SVs detected in H20D and H20H using nine long-read SV calling strategies (Supplementary Data 14). Performance of H20H varied greatly depending on whether

haploid references were used, with details in the Supplementary Notes (Fig. 3d, e and Supplementary Data 14). Also, H20H had lower accuracy: the F1 score declined from 0.75 to 0.71 across all tools, and from 0.79 to 0.68 when excluding haploid strategies, with precision dropping sharply (0.82 to 0.58) despite a slight increase in recall (0.76–0.82) (Fig. 3f, g and Supplementary Data 15). To validate these results, we analyzed 10 Jersey HiFi samples and built haplotype-resolved (J10D) and primary assembly-based (J10H) pangenome graphs (See Supplementary Data 6 and the Supplementary Notes). J10H was smaller and less complex, with fewer nodes, edges, and variants than J10D

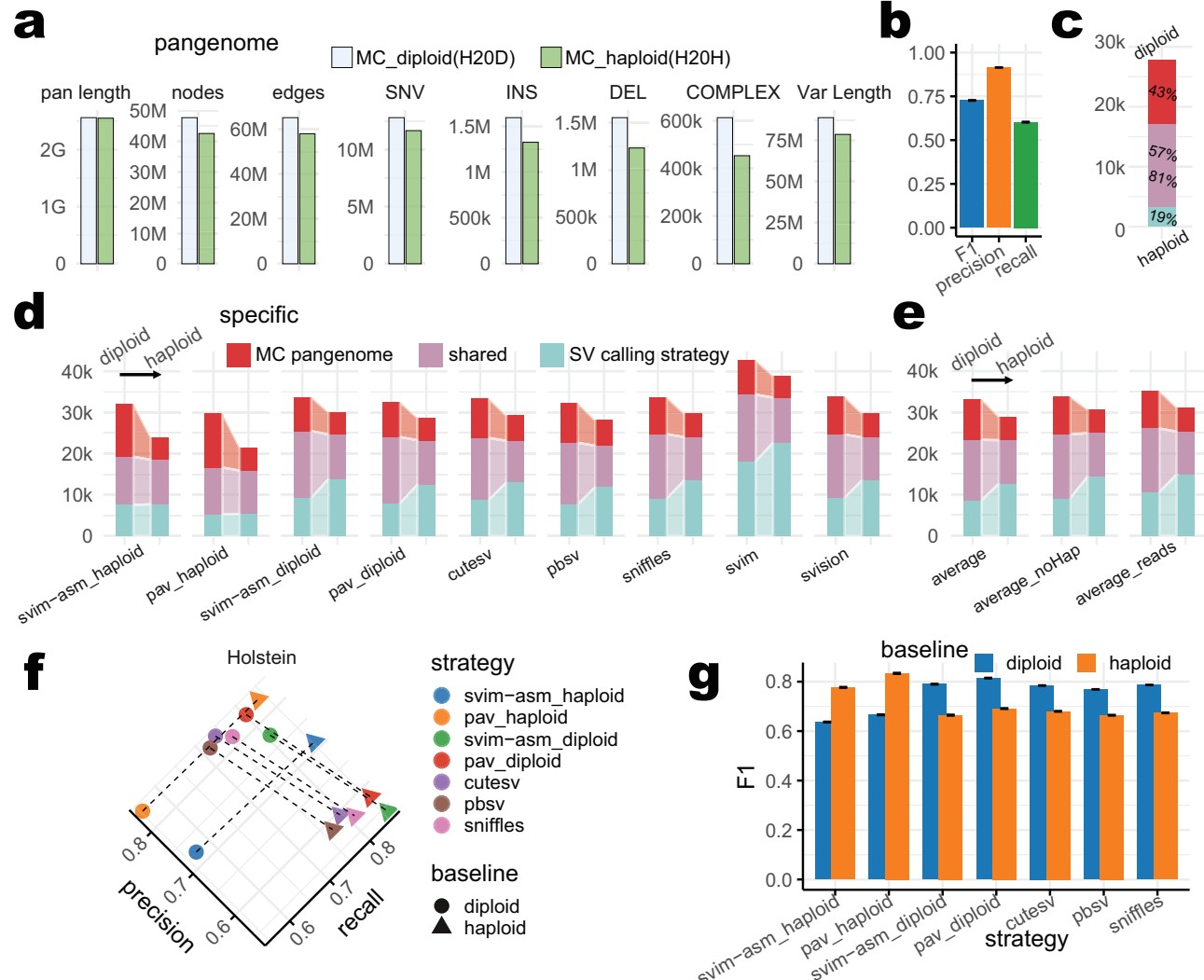

**Fig. 3 | Comparison of SV calling strategies using pangenome graphs constructed from diploid phased assemblies (H20D) versus the haploid primary assemblies (H20H).** **a** Pangenome graph metrics: Comparison of pangenome sizes, counts of nodes, edges, 4 variation types (SNV, INS, DEL, and COMPLEX), and the total lengths of all variations between the H20D and H20H for 20 Holstein samples. **b** F1 score, precision rate, and recall of H20H were evaluated against H20D. Bars indicate mean ± SE (standard error) with n = 20. **c** The proportion of unique and shared SVs between H20D and H20H. Percentages denote unique/shared proportions for H20D (upper/middle) and shared/unique proportions for H20H (middle/lower). **d** Count changes of unique and shared SVs for each of the 9 strategies when compared against H20D or H20H. **e** Average count changes of unique and shared SVs across all strategies, all strategies excluding SVIM-asm_Haploid and PAV_Haploid, and 5 read mapping-based strategies as compared to H20D or H20H. **f** Precision and recall rates for 7 strategies as compared to H20D or H20H. **g** F1 scores for 7 strategies with H20D or H20H as the baseline. Bars indicate mean ± SE (standard error) with n = 20.

(Supplementary Fig. 7a–c, Supplementary Data 7 and Supplementary Data 13). Consistent with the Holstein findings, J10D identified more SVs (22.53 k vs. 16 25 k), shared a higher proportion with other tools (60.48% vs. 49.38% in Supplementary Data 16), and had stronger concordance with read mapping-based callers (F1 = 0.77 vs. 0.69), mainly due to higher precision (0.80 vs. 0.61) (Supplementary Fig. 7d and Supplementary Data 17). All these findings demonstrate that diploid models outperform haploid models for pangenome construction, detecting more SVs, aligning better with read mapping-based strategies, and providing fully phased variation.

**Single-breed pangenome with the diploid model enhances SV genotyping and phasing in difficult genomic regions**

We compared diploid (H20D) and haploid (H20H) pangenome models by analyzing two SVs in Holstein Sample 4890: a 391 bp deletion (chr1:1,579,558) detected by both the diploid model (MC) and other tools, and a 132 bp insertion (chr1:1,579,847) identified only by the diploid model, as it did not meet the 90% overlap threshold with others (Fig. 4a, b, Supplementary Data 18). Both variants were present on chr1 haplotypes (1|0 for the insertion, 0|1 for the deletion). H20D captured both SVs, whereas H20H missed the insertion (Supplementary Data 19). Excluding haploid models, read mapping- and diploid assembly-based tools detected both SVs. Located in a tandem repeat region (TGTGTG/GTGTGT), these SVs show mapping variability (Supplementary Fig. 8). The insertion exhibited more positional and length differences across tools, with Sniffles and diploid models of PAV and SVIM-asm matching the MC length but shifting by 33–44 bp (Fig. 4c and Supplementary Data 20), while SVision, cuteSV, and pbsv reported differing sizes and locations. In H20D, the INS and DEL corresponded to nodes spanning 1,072,043–1,072,086 (Fig. 4c). The INS followed the path >1072056 >1072057 >1072058, and the DEL >1072043 < 1072044 >1072086, classified as alleles 3 and 4 among five alleles (one reference, four alternatives) (Fig. 4c–e and Supplementary Data 19). Among 20 samples, two (4245, 4450) were missing genotypes in this

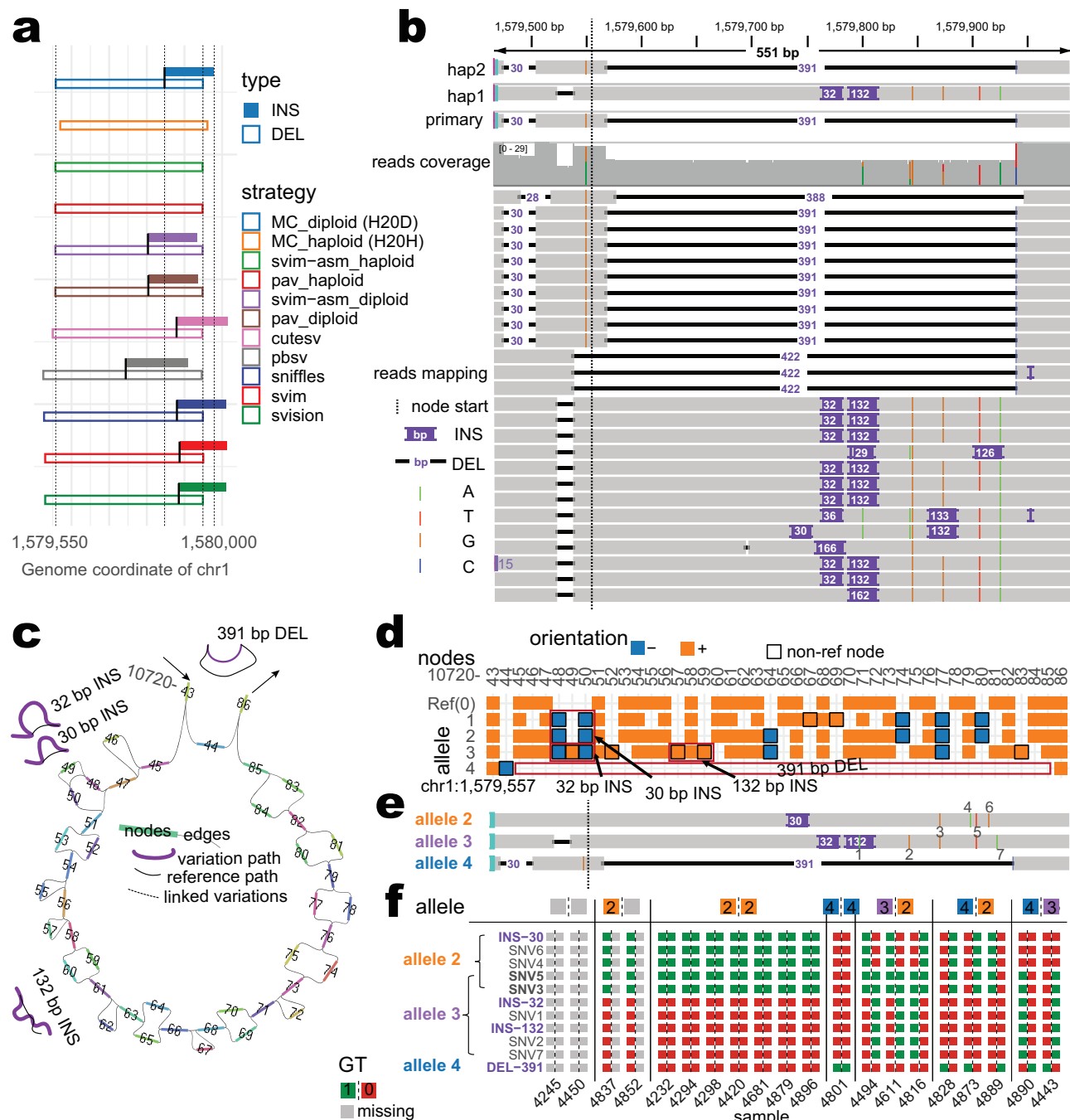

**Fig. 4 | An example of DEL and INS classifications from H20D, H20H, and 9 long-reads calling strategies. a** Genome coordinates of the INS and DEL identified by 11 strategies. **b** Integrative Genomics Viewer (IGV) screenshot of chr1:1,579,000–1,580,500 for Sample 4890, showing BAM alignments from both assembly and read mapping. The genome tracks, from top to bottom, include haplotype 1, haplotype 2, and the primary assembly, followed by HiFi read coverage in this region and individual HiFi read mappings. **c** BandageNG plot for the pan-genome nodes starting at 1,072,043 and ending at 1,072,086. Three variations classified by walks against the ARS_UCD2.0 reference are shown inside the circle.

**d** DEL and INS classifications from pangenome nodes 1,072,043 to 1,072,086 at the genome coordinate chr1:1,579,557. Colored rectangles denote node orientations, with black rectangles highlighting non-reference nodes and red rectangles marking a 132 bp INS and a 391 bp DEL. **e** The IGV visualization of the three ancestral alleles collected from the 16 samples. **f** Allele combinations and variant genotypes for the three alleles in 20 samples. The allele combinations of the sample groups are shown at the top. SNV3 and SNV5 are shared between alleles 2 and 3. The red box indicates genotype 1 (present), while the green box indicates genotype 0 (absent), and the gray box represents missing data.

bubble, and two (4837, 4852) were only partially genotyped (Supplementary Data 20). Partial genotyping often arose from missing sequences in haploid assemblies or conflicts caused by nearly identical sequences in terminal regions of chr1 (Supplementary Figs. 9 and 10). For example, in Sample 4450, two highly similar sequences enabled genotyping in the haploid model H20H, while in Sample 4245, minor sequence differences caused genotyping failure in both haploid and

diploid graphs. For details, please see the Supplementary Notes. For allele classification, the allele shared across all three groups was designated as the putative ancestral allele. Allele 1 differed from allele 2 by a single SNV (1:1,579,792:T > G) and was merged into allele 2 for downstream analyses. The remaining 16 samples' genotypes were represented by three alleles: allele 2 (30 bp insertion + four SNVs), allele 3 (linked 32 bp and 132 bp insertions + four SNVs, two shared with

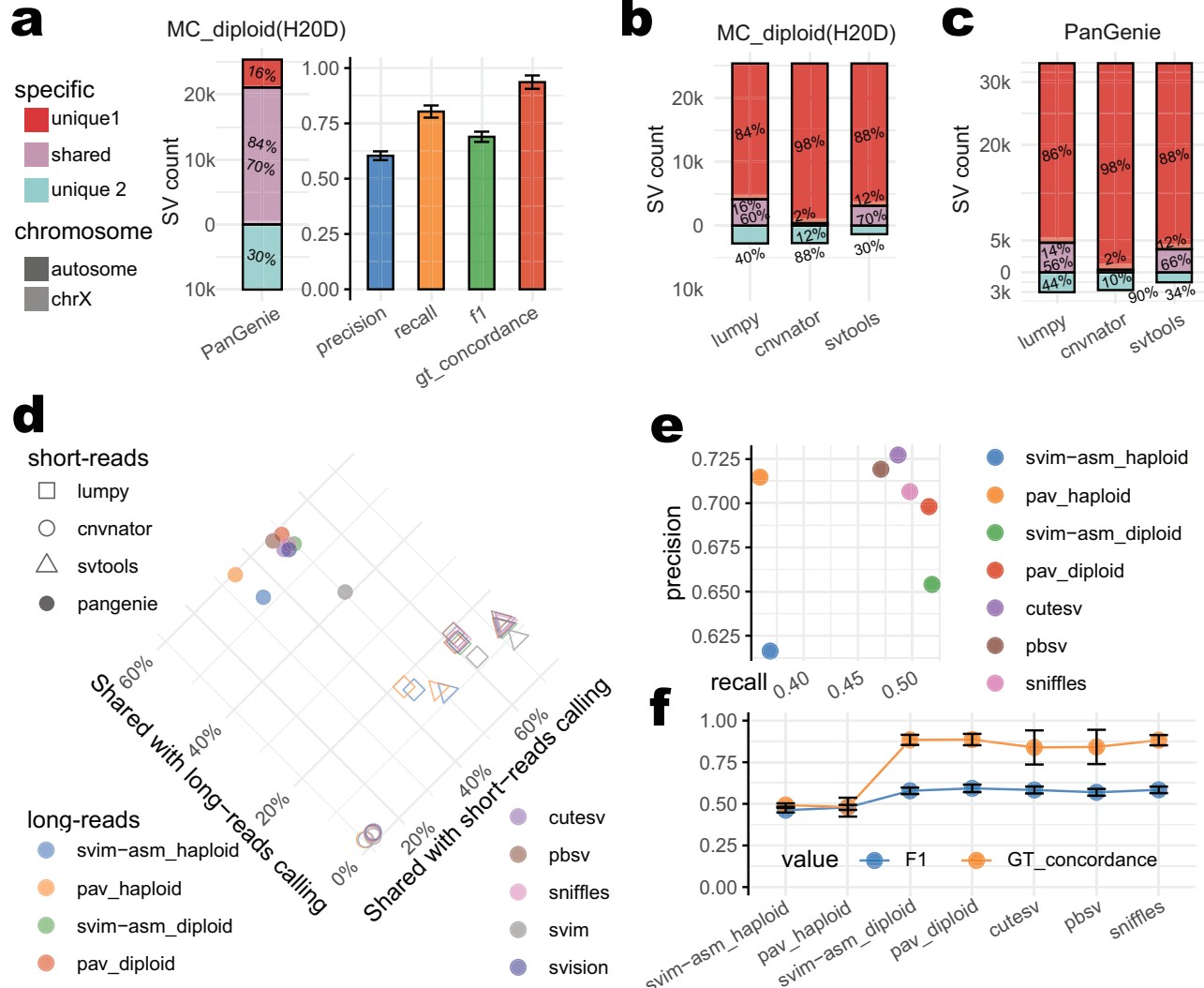

**Fig. 5 | Comparison of SV calling using pangenome PanGenie and WGS strategies. a** Proportion of unique and shared SVs between PanGenie and the H20D pangenome-based strategy. F1 score, precision, and recall of the PanGenie strategy were evaluated against H20D. Bars indicate mean ± SE (standard error) with n = 20. **b** Proportion of unique and shared SVs between 3 short-read SV calling strategies (Lumpy, CNVnator, and SVtools) and the H20D pangenome-based strategy. **c** The proportion of unique and shared SVs between the same 3 short-read SV calling strategies and PanGenie. **d** The proportion of shared SVs between the PanGenie and 3 short-read SV calling strategies versus 9 long-read SV calling strategies (SVIM-asm_Diploid, SVIM-asm_Haploid, PAV_Diploid, PAV_Haploid, cuteSV, pbsv, Sniffles, SVIM, and SVision). **e** Precision and recall rates, as well as **f** genotype concordance for 7 strategies as compared to PanGenie. Bars indicate mean ± SE (standard error) with n = 20.

allele 2), and allele 4 (319 bp deletion) (Fig. 4e). In Samples 4890 and 4443, the 132 bp INS and 391 bp DEL arose from alleles 3 and 4 (4 | 3), while other samples carried 3 | 2 (4494, 4611, 4816), 4 | 2 (4828, 4873, 4889), 2 | 2 (4232, 4294, 4298, 4420, 4681, 4879, 4896), or 4 | 4 (4801) genotypes (Fig. 4f and Supplementary Figs. 11 and 12). These results demonstrate that diploid pangenomes outperform haploid models by detecting more SVs, aligning better with read mapping-based strategies, and resolving complex allelic variation, including both insertions and deletions at the same locus.

**Pangenome improves SV genotyping in short reads**

To evaluate the advantages of using a pangenome reference for SV genotyping in short-read sequencing data, we generated and analyzed whole genome sequencing (WGS) for the same 20 Holstein samples, with coverage ranging from 30.3× to 40.0× (Supplementary Data 21). Using PanGenie[42], a k-mer-based short-read genotyping tool that incorporates fully assembled haplotypes, we genotyped on average ~31.95k autosomal SVs per sample with H20D

as a reference. A high proportion of shared SVs was observed between H20D and PanGenie (83.54% for H20D and 69.79% for PanGenie), underscoring the value of a pangenome reference in improving SV genotyping accuracy over a standard linear reference. When applying a 90% overlap threshold, PanGenie reported that an average of 30.21% (~9.65k, up to 31.95k) of SV calls per individual were unique, as compared to the H20D pangenome SVs, achieving an F1 score of 0.69 and a genotype concordance of 0.93 (Fig. 5a).

To further evaluate the benefits of using a pangenome graph as a reference for short-read SV genotyping, we compared H20D and PanGenie-guided calls with 3 short-read SV detection tools: Lumpy, CNVnator, and SVtools[66,67] (Fig. 5b, c). All short-read tools, including Lumpy, CNVnator, and SVtools exhibited low proportions of shared SVs (~16%, 2%, and 12%) with H20D, respectively (Fig. 5b). PanGenie detected significantly more SVs overall (31.95k vs. 6.56k, 2.65k, and 4.40k for SVtools, Lumpy, and CNVnator, respectively) (Fig. 5c and Supplementary Data 22). Additionally, only 14%, 2%, and 12% of

PanGenie's calls overlapped with 56% Lumpy, 10% CNVnator, and 66% SVtools calls, respectively (Fig. 5c).

Using long-read SV calling methods as a benchmark, PanGenie achieved a significantly higher average shared rate of 55.61% across 9 strategies, again outperforming the 3 short-read methods: SVtools (12.73%), Lumpy (17.14%), and CNVnator (1.64%) (Fig. 5d). Among the 9 long-read strategies, the PAV diploid model demonstrated the highest F1 score (0.60) and recall rate (0.51) with PanGenie, while cuteSV and pbsv achieved the highest precision rate (both at 0.73). Genotype concordance was highest for the PAV diploid model, pbsv, and cuteSV (all near 0.89), followed closely by SVIM-asm_Diploid and Sniffles (both near 0.88) when compared with PanGenie (Fig. 5e, f).

Given that the same samples were used for both pangenome construction and short-read variation genotyping, there is a potential for statistical bias, which could overestimate the consistency between the H20D pangenome dataset and the PanGenie short-read genotyping. To address this concern, we also tested the previously built Jersey single-breed pangenome using the diploid model from 10 Jersey HiFi samples (J10D, see Methods). We did not directly compare PanGenie results for the 20 Holsteins using H20D vs. J10D, as differences in sample size and pangenome structure would confound results. Instead, nine alternative strategies were used as background references to provide a fairer assessment of consistency rates. Using J10D as a reference, PanGenie genotyped an average of 22.85k autosomal SVs in the 20 Holstein WGS samples, confirming higher SV detection consistency vs. traditional short-read methods. PanGenie reported an average of 10,086 SVs shared with the 9 strategies, achieving an average shared rate of 44.17%, significantly higher than Lumpy (17.14%, 3.91k), CNVnator (1.64%, 0.37k), and SVtools (12.73%, 2.91k) (Supplementary Data 23). We also genotyped 10 Jersey WGS data (~31.42×, labeled as PanGenie-Jer) using the H20D pangenome graph as the reference. On average, 27.28k autosomal SVs were detected, with 11.53k (42.26%) shared with the 9 strategies. This shared rate was 2 times higher than Lumpy (3.73k, 17.00%), 3 times higher than CNVnator (0.37k, 1.73%), and 40 times higher than SVtools (2.82k, 12.46%) (Supplementary Data 24).

## SNV consistency evaluation with the pangenome strategy

In addition to SVs, we assessed SNV consistency across 20 Holstein samples using HiFi and WGS data with RTG Tools. We compared three SNV calling methods, Clair3 on HiFi (C3(HiFi)), DeepVariant on WGS (DV(WGS)), and a hybrid HiFi+WGS DeepVariant model (HB(HiFi+WGS) or HB), alongside three pangenome-based approaches: Minigraph-Cactus with *vg deconstruct* (MC(HiFi)), PanGenie on WGS with Holstein H20D (PG(WGS/H20D)), and PanGenie on WGS with Jersey J10D (PG(J10D)). On average, the hybrid DeepVariant model HB reported the most SNVs per sample (6.99 M), closely followed by C3(HiFi) (6.96 M), with C3(HiFi) achieving the highest F1, precision, and sensitivity (~0.97) relative to HB (Fig. 6a, b and Supplementary Data 25), although HB results do not represent the gold-standard truth sets. MC(HiFi) reported 6.38 M SNVs, comparable to DV(WGS) (6.46 M), with similar F1 (~0.90), sensitivity (~0.94), and precision (0.86–0.87). PG(J10D) and PG(WGS/H20D) retained high sensitivity (0.93 vs. 0.91), but PG(J10D) identified fewer SNVs (5.14 M, 69.83% of HB) and had lower precision (0.67 vs. 0.84), highlighting limitations when using a pangenome from another breed. In SV regions, C3(HiFi) detected slightly more SNVs than HB (0.29 M vs. 0.28 M) and PG(WGS/H20D) captured 10.1% more SNVs in insertions than MC(HiFi) (0.12 M vs. 0.11 M) (Fig. 6c). MC(HiFi) SNVs had higher F1 with HB (0.80) than PG(WGS/H20D) (0.73) or DV(WGS) (0.75). WGS-based approaches (PG and DV) showed high SV-region consistency (F1 = 0.87), though DEL and COMPLEX regions had lower SNV consistency than INS regions across tools (Fig. 6d). In repetitive regions, C3(HiFi) reported more SNVs than HB, except in satellites, where HB detected over twice as many SNVs (46.84k vs. 14.06k), indicating higher sensitivity, although

accuracy was not directly assessed. Among pangenome methods, PG(WGS/H20D) and DV(WGS) showed similar performance in satellites (2.89k vs. 2.56k SNVs), while PG(J10D) detected slightly more (3.34k) (Fig. 6g and Supplementary Data 25). Overall, these results demonstrate that pangenome-based approaches, particularly those with phased haploid references, provide SNV genotyping with high consistency across methods in both SV and repetitive regions, whereas using a pangenome from a different breed reduces precision.

## Pangenome SV-based GWAS in 173 WGS samples

To investigate the role of SVs in complex cattle traits, we used PanGeine to genotype H20D pangenome-derived SVs in 173 Holstein cattle, which have been reported before[68,69]. A total of 61,249 SVs with a minor allele frequency (MAF) ≥ 0.05 were analyzed, including 16,261 DELs, 26,284 INSs, and 18,704 COMPLEX events. We performed GWAS analyses for 46 traits of 5 main trait types, including Conformation and Type (CT, n = 24), Feed Efficiency (FE, n = 1), Fertility and Calving (FC, n = 8), Longevity and Health (LH, n = 10), and Production and Yield (PY, n = 4). Using EMMAX (v20120210) and a mixed linear model, we detected a total of 196 significant SV-trait associations at a genome-wide significance threshold of $8.58 \times 10^{-7}$ (Fig. 7a, Supplementary Fig. 13, and Supplementary Data 26). Of these, 135 SVs (30 DEL, 69 INS, and 36 COMPLEX events) showed significant association with 42 traits. There were 22 traits significantly associated with multiple SVs (n > 2), such as cattle production and yield traits Protein (Pro) and Fat (FAT) associated with 14 SVs (3 DEL, 7 INS, and 4 COMPLEX events) and 13 SVs (3 DEL, 8 INS, and 2 COMPLEX events), respectively. Fore udder attachment (FUA) was linked to 12 SVs (4 DEL, 6 INS, and 2 COMPLEX events), and Heifer Livability (HLV) was associated with 11 SVs (3 DEL, 6 INS, and 2 COMPLEX events), respectively. These findings highlight that SVs not only are readily detectable but also often co-locate with GWAS signals from SNVs, indicating their potential functional impact on economically important traits in Holstein cattle.

To further explore the biological relevance of these SVs, we annotated those significant SV regions and identified 49 protein-coding genes, 19 long non-coding RNAs (lncRNAs), and 7 pseudogenes affected by 69 SVs (Supplementary Data 27). Of them, 9 SVs overlapped 9 protein-coding genes, affecting 18 coding sequences (CDS). We also annotated those significant regions using the promoter (Strongly active promoters/transcripts) and enhancer (Strong active enhancers) of chromatin states in 8 tissues. A total of 28 promoters and 29 enhancers of 14 and 13 genes were annotated by 12 and 13 SVs, respectively (Supplementary Data 28). These 57 regulatory elements were located in 12 gene bodies, including *EPPK1*, *CERS6*, *TG*, *LOC100847180*, *RTN4*, *FOXK2*, *NOS3*, *PSEN2*, *DNAH5*, *FBXL2*, *DNM2*, and *CNOT6L*, as well as 10 gene flanking regions, including *HPCAL1*, *EFR3A*, *LOC100336734*, *LOC100847180*, *BHMT*, *HS3ST4*, *LOC112441461*, *LY6K*, *CYP11B1*, and *MATN3*.

We define SV-related genes as those that overlap or lie near SVs and may be affected in dosage, regulation, or structure. Among these SV-related genes, 21 were linked to promoter or enhancer regions, suggesting potential regulatory effects. We use 2 examples for illustration of how SV potentially affects gene function. One is a 6179 bp DEL (Chr11:78819207-78825386:DEL, frequency = 59) that was significantly associated with the Stature of cattle conformation and type (ST, $P = 2.06 \times 10^{-7}$). This DEL covered 10.65% length of the *MATN3* (Matrilin-3) gene, which is related to premature chondrocyte development and affects bone mineral density[70]. Additionally, this deletion not only overlaps the upstream enhancers of the *MATN3* gene in the liver and lung but also entirely covers the first exon of the *MATN3* gene, which is shared by 3 transcripts (Supplementary Data 29 and 30). The 6178 bp DEL potentially affects the *MATN3* gene's function at its critical region. These findings suggest that this SV may disrupt MATN3 function at a critical regulatory site, potentially affecting bone development and growth in Holstein cattle. The other example is a 1548 bp INS

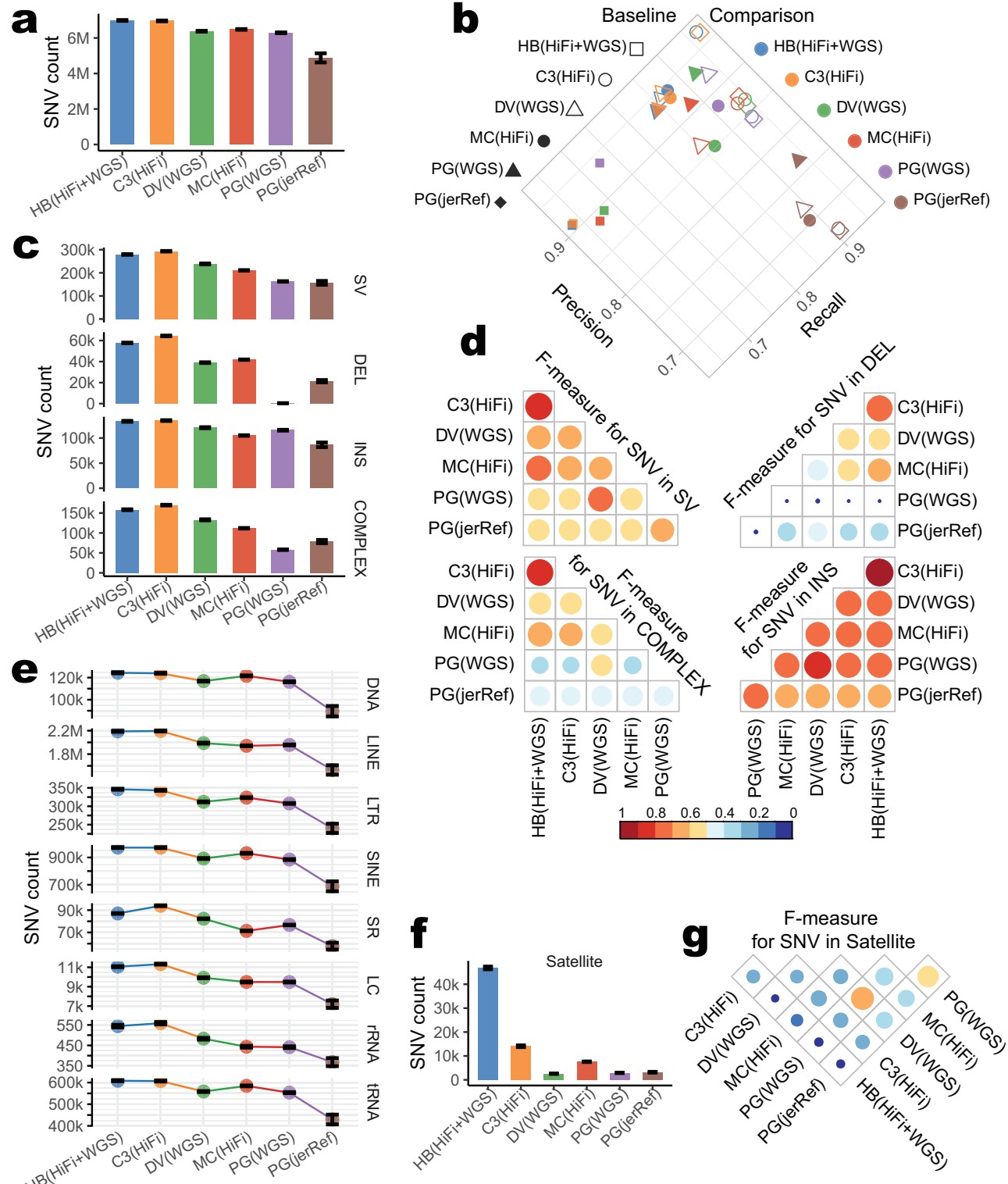

**Fig. 6 | Consistency evaluation across SNV callers by RTG Tools. a** Comparison of SNV counts reported by a hybrid DeepVariant model (HB(HiFi+WGS)), Clair3 on HiFi data (C3(HiFi)), DeepVariant on WGS data (DV(WGS)), Minigraph-Cactus pipeline for pangenome graph generating and *vg deconstruct* for the VCF exporting on HiFi data (MC(HiFi)), PanGenie for Holstein WGS with H2OD as the reference (PG(WGS)), and PanGenie for Holstein WGS with the Jersey pangenome graph (J1OD) as the reference (PG(JerRef)). Bars indicate mean ± SE (standard error) with n = 20. **b** Precision and recall rates for each caller against other callers. Base: Baselines are denoted by 6 symbols; Comparison: Comparisons are made by 6 colors. **c** SNV counts reported by each caller within each SV type or all SV types. Bars indicate mean ± SE (standard error) with n = 20. **d** F-measure values for SNVs detected by each caller within each SV type or all SV types. **e** SNV counts reported by each caller within repetitive regions. **f** SNV counts reported by each caller within satellite regions. Bars indicate mean ± SE (standard error) with n = 20. **g** F-measure values for SNVs reported by each caller within satellite regions. Where boxplots were used, the center line represents the median for 20 independent individuals (n = 20), box limits depict the interquartile range (IQR), and whiskers extend to 1.5×IQR; all data points were included, and no outliers were removed.

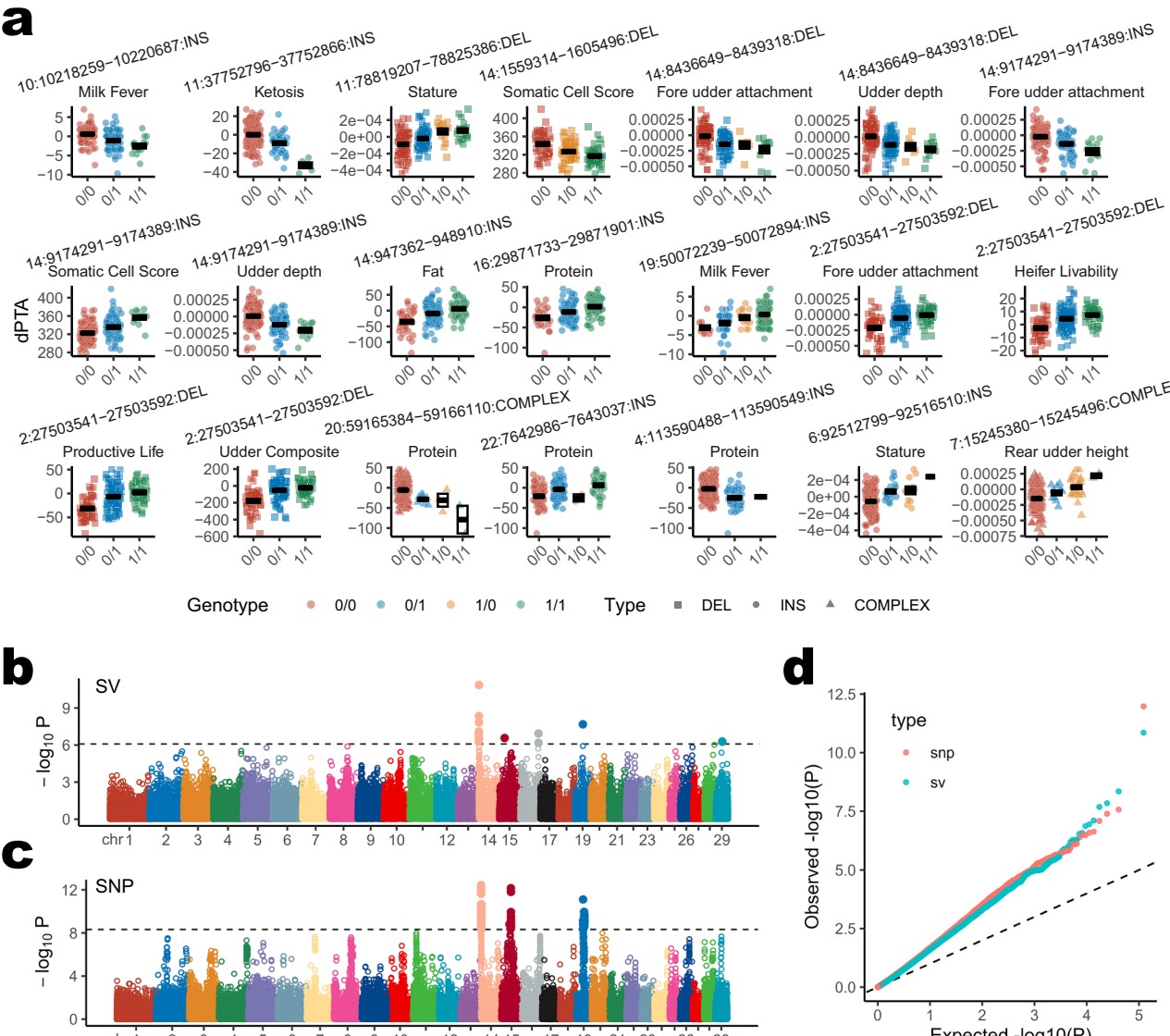

**Fig. 7 | Pangenome variation-based GWAS for cattle phenotype traits.**
**a** Visualization of genotype-phenotype associations for various traits (e.g., milk fever, ketosis, stature, somatic cell score, etc.). SV positions and types are shown at the top. The X-axis represents genotypes with different colors (0/0: homozygous reference 0/1 and 1/0,: heterozygous, 1/1: homozygous alternative). The Y-axis represents phenotypic trait values (e.g., dPTA). For the boxplots, the center line represents the average phenotypic dPTA value for the 173 samples divided by the

genotypes, and the bounds of the box show mean ± SE (standard error). Statistics are based on 173 biological replicates (n = 173), each representing one independent Holstein individual. **b** Manhattan plot of GWAS −$\log_{10} P$ values for SVs associated with fat with EMMAX (v20120210) using a mixed linear model. **c** Manhattan plot showing GWAS −$\log_{10} P$ values for SNPs associated with fat using EMMAX. Dashed lines indicate significance thresholds. **d** Q-Q plot for GWAS results based on SNV and SV.

(Chr14:947362-948910, frequency = 86), which was significantly associated with fat percentage (FAT, $P = 1.27 \times 10^{-7}$). This insertion overlaps the CDS, exon, and first-exon promoters of the *EPPK1* gene in adipose and cortex tissues (Chr14:947,200−947,400).

Comparisons of SV-based and SNP-based GWAS revealed similar patterns of significance, with overlapping peaks in both analyses (Fig. 7b, c). However, a higher proportion of SVs than SNPs reached genome-wide significance at $P < 10^{-4}$ (Fig. 7d). This suggests that SVs capture additional genetic variation that may be missed in traditional SNP-based GWAS, reinforcing the importance of integrating SVs into genomic selection models for livestock breeding.

## Discussion
### Study overview
Our study demonstrates the value of pangenomic approaches in capturing the genetic diversity underlying key traits in livestock, focusing

on Holstein cattle. By constructing within-breed and crossbreed pangenome graphs, we provide a more comprehensive representation of SV beyond what a single-reference genome can offer. The Holstein-specific pangenome graph (H2OD) enabled us to identify SVs unique to the breed, offering insights into the genetic basis of economically significant dairy production traits, such as body weight, milk yield, and fertility. These findings underscore the limitations of relying solely on a single-reference genome for livestock genetics, as traditional approaches risk overlooking crucial within-breed and inter-breed variations that influence economically important traits.

### Enhanced SV detection and genotyping accuracy
The Holstein-specific pangenome graph exhibited fewer non-reference nodes and edges than the crossbreed pangenome graph (M13H), reflecting the breed's intensive breeding history and underscoring the advantages of using a pangenome reference for SV genotyping in

short-read sequencing data. By leveraging PanGenie, we demonstrated that SV detection using a within-breed pangenome reference improves genotyping accuracy and consistency relative to traditional short-read mapping methods. Importantly, PanGenie guided by H20D outperformed CNVnator, Lumpy, and SVtools in both detection rate and genotype concordance. These findings demonstrate that using a pangenome reference significantly enhances SV detection, achieving performance comparable to long-read mapping-based methods. The improved consistency between PanGenie and long-read SV calling tools further highlights the benefits of pangenome-guided genotyping, particularly in capturing COMPLEX variants that may be misclassified or missed by conventional reference-based approaches.

## SVs play a significant role in phenotypic diversity

Our GWAS further underscores the importance of SVs in cattle phenotypic variation, identifying over 196 significant gene-trait associations across 46 traits. The results highlight how SVs contribute to phenotypic complexity beyond SNP-based analyses, reinforcing the need to integrate SVs into future genomic selection models. Among the detected associations, SVs linked to body weight, stature, milk production, and fertility were particularly notable. For example, a 6179 bp deletion on chromosome 11 associated with stature overlaps the *MATN3* gene, which influences bone development. Given that stature influences both body mass and feed efficiency, this deletion may serve as a candidate marker for future genetic selection strategies. Similarly, a 1548 bp insertion on chromosome 14 overlapping EPPK1 was significantly associated with milk fat percentage, suggesting that SVs in regulatory or coding regions can influence dairy production. Given EPPK1's role in epithelial integrity and cellular adhesion, this insertion may affect fat deposition or metabolism. These findings highlight the biological relevance of SVs in cattle genetics and demonstrate how pangenome-based SV detection enhances trait association analyses, uncovering variants that influence economically significant traits and providing opportunities for genetic improvement in Holstein cattle.

## Implications for livestock breeding and genetic improvement

Our findings highlight the future potential of pangenome-based analyses in livestock breeding. Detecting within-breed and crossbreed SVs provides a foundation for potentially improving genomic selection, as it may help breeders identify alleles associated with economically valuable traits. Importantly, the Holstein-specific pangenome graph enhances the ability to track within-breed genetic variation, making it a valuable tool for breed improvement programs. Conversely, the crossbreed pangenome graph facilitates the identification of core SVs shared across breeds, providing insights into evolutionary conservation and genomic stability. These findings suggest that integrating pangenomic resources into breeding programs could improve genetic prediction models, leading to higher milk yield, enhanced fertility, and improved resilience to environmental stressors. Furthermore, the inclusion of SVs in genomic selection models may allow for more precise genetic evaluations, reducing inbreeding risks while maintaining genetic diversity in elite dairy cattle populations. Given the increasing reliance on genomic selection in dairy breeding programs, incorporating high-confidence SVs alongside SNP markers could enhance trait heritability estimates, ultimately leading to more efficient and productive cattle populations.

## Limitations and future directions

While our study provides valuable insights, several limitations must be acknowledged. First, the sample size remains a constraint, particularly in detecting rare SVs with low allele frequencies. Future studies incorporating larger datasets and additional cattle breeds will be essential to validate and expand upon these findings. Second, our study focused on SVs larger than 50 base pairs, meaning that smaller

SVs, such as micro-indels, were not analyzed. These smaller variants may still contribute to phenotypic variation and should be explored in future research. Additionally, while our functional annotations identified protein-coding genes, lncRNAs, and regulatory elements associated with significant SVs, further experimental validation is necessary to confirm their biological effects. Future studies integrating multi-omics approaches, such as RNA-seq and epigenetic data from FarmGTEx[71], could provide deeper insights into how SVs influence gene expression. Moreover, as long-read sequencing technologies advance, future research should leverage deeper sequencing and phased assemblies to better resolve complex SVs and their impact on traits. Expanding pangenomic analyses across additional breeds and species, including dairy and beef cattle, swine, and poultry, will not only broaden our understanding of SVs in domesticated animals but also address key challenges of SV assessment across diverse genome architectures and evolutionary histories, guiding future studies in diverse populations.

Despite these limitations, our study underscores the power of pangenomic approaches in capturing diverse SV landscapes in cattle. It provides validation to guide the design of future studies, particularly those aimed at improving the efficiency of SV genotyping in large whole-genome sequencing cohorts. Our findings highlight the importance of including SVs in genetic studies, as they represent a major source of genomic variation that traditional SNP-based analyses may overlook. As sequencing technologies advance and computational methods improve, pangenomics will play an increasingly critical role in precision breeding and sustainable agriculture.

## Methods
### Data collection and processing

All samples were from females/cows, as the study focused on dairy cattle. We retrieved HiFi sequencing data for 20 Holstein and 8 Jersey cows from a previous study[47]. We generated similar HiFi data for 2 additional Jersey cattle. The data had an average read length of 13–23 kb and coverage of 23.2–33.5× relative to the ARS-UCD v2.0 reference genome. Briefly, for de novo assembly, we utilized the Hifiasm (v.0.18.5) assembler[72], resulting in the generation of 30 primary haploid assemblies and 60 phased diploid assemblies. The average total primary genome lengths of 3.26 and 3.16 Gb, with average contig N50s of 72.98 Mb and 56.80 Mb for the Holstein and Jersey, respectively. According to BUSCO (v.5.4.3)[73], the completeness of mammalian universal single-copy orthologs in the Holstein and Jersey genome assemblies was found to be 94.99% and 95.88%, respectively, which was consistent with the 95.80% seen in previously analyzed cattle genomes[47]. Similarly, the duplication rates of 2.14% and 2.29% in the assemblies are consistent with the typical 2.00% range observed in Hereford genomes. BUSCO scores for the phased assemblies were slightly lower but still consistently >90% complete. The quality value (QV) of the assemblies assessed by the Inspector (v.1.0.2)[74] indicated high QV values for all the primary and phased assemblies (average over 44). According to Rhie et al.[75], all these assemblies were therefore considered as high-quality under the VGP-2020 standards. To improve contig phasing accuracy, Hi-C data were generated for 2 selected Holstein individuals and assembled in Hifiasm's Hi-C integration mode.

We also downloaded 8 publicly available primary assemblies with Contig N50 values exceeding 1 Mb (GCA_021234555.1, GCA_021347905.1, GCA_028973685.2, GCA_034097375.1, GCA_039881175.1, GCA_905123515.1, GCA_905123885.1, and GCA_947034695.1) from NCBI, along with 5 additional HiFi assemblies (H_hifiasm, N_hifiasm, O_hifiasm, B_hifiasm, and P_hifiasm) representing various breeds, as reported in Leonard et al.[55]. In total, 13 assemblies from different breeds were analyzed, including 9 *Bos taurus* breeds (Jersey, Holstein, Charolais, Braunvieh, Brown Swiss, Piedmontese, African N'Dama, and Hanwoo), 2 *Bos indicus* breeds (African Ankole, and Nellore), and 2 crossbred cattle (Yunling and Mongolian) (Supplementary Data 1). The

GFA files of Holstein samples were converted to FASTA format using gfatools (v.0.4).

To compare the performance of SV calling across multiple approaches, the 20 Holstein and 10 Jersey genomes were also sequenced using whole-genome sequencing (WGS) (Illumina NextSeq 2000 Systems) with a coverage range of 30.3–40.0× against ARS-UCD v2.0.

## Pangenome construction

For pangenome graph construction, we used the Minigraph-Cactus (MC) pipeline (v2.4.2) with the ARS-UCD v2.0 genome as the backbone[38]. The MC pipeline combines Minigraph, which constructs a variation graph from multiple genomes, with Cactus, which enables accurate whole-genome multiple sequence alignment (WGA). We created 5 pangenome graphs using this pipeline: (1) H20D with 20 phased diploid assemblies from Holstein; (2) M13H with 13 primary haploid assemblies from diverse breeds; (3) H20H with 20 primary haploid assemblies from Holstein; (4) J10D with 10 phased diploid assemblies from Jersey; (5) J10H with 10 primary haploid assemblies from Jersey. MC parameters included "cactus-pangenome --binariesMode local --reference ARS_UCD_v2.0 --giraffe clip filter --gbz clip filter full --gfa clip filter full --vcf --permissiveContigFilter --haplo --chrom-vg clip filter --chrom-og full --viz". The "--permissiveContigFilter" parameter was used to enhance sensitivity in regions with fragmented or highly diverse contigs. For example, to construct H20D, we ran the MC pipeline using 64 CPU cores, with its sub-tasks configured to use 32 cores and approximately 8 GB of RAM per core. Under this configuration, the entire MC workflow required about one week to complete. To compare and assess the pangenome graphs, we used Panacus (v0.2.4) to calculate various statistics, including node, edge, and path counts for each graph (https://doi.org/10.1101/2024.06.11.598418).

## SV calling for HiFi data
### Pangenome-based strategy

**Minigraph-Cactus (MC).** The Minigraph-Cactus pipeline was used for SV calling with the VCF exporter in vg (utilizing *vg deconstruct*) to generate variant calls. Each site in the graph corresponding to a "snarl" (a variant region) was outputted, with the haplotype index (GBWT) identifying haplotypes traversing each site to produce phased genotypes. Nested snarls were included for each allele. To classify SVs and convert multi-allelic SVs into biallelic format, we used a script from the PanGenie (v3.1.0) workgroup (https://github.com/eblerjana/genotyping-pipelines/tree/main/prepare-vcf-MC/workflow/scripts/annotate_vcf.py) that classifies SVs into 3 types: INS (insertion), DEL (deletion), and COMPLEX. Here, "COMPLEX" refers to variant alleles within regions that contain complex bubbles (with more than 2 branches) in the pangenome graph[60].

## Assembly-based strategies

**Minimap2.** We aligned assembled FASTA sequences to the ARS-UCD v2.0 reference genome using Minimap2 (v2.24) with the following parameters: -x asm20 -m 10000 -z 10000,50 -r 50000 --end-bonus=100 --secondary=no -O 5,56 -E 4,1 -B 5 -a --eqx -Y.

## Phased Assembly Variant Caller (PAV)

We applied PAV (v2.3.4) to discover variants with default parameters for both diploid (hap1 and hap2 assemblies) and haploid (primary assembly) models[42].

## SVIM-asm

We separately run SVIM-asm (v.1.0.2) with diploid and haploid models for SV detection[61]. The options were set to --tandem_duplications_as_insertions --interspersed_duplications_as_insertions --min_sv_size 50 --max_sv_size 200000, thus restricting output SVs to lengths between 50 bp and 200 kb.

## Mapping-based strategies

**Read Mapping.** We mapped HiFi reads (filtered by fastp v0.23.4 to retain reads ≥1 kb) to the ARS-UCD v2.0 reference genome using Minimap2 (v2.24) with parameters '-L --cs -H -Y --MD'. We then removed alignments with mapping quality <30 using SAMtools (v1.20) with '-q 30'. chrY and unplaced chromosomes were excluded from this analysis. We then applied the following SV detection tools:

## SVIM

We employed SVIM (v1.4.2) for mapping-based SV calling, with parameters '--segment_gap_tolerance 10 --segment_overlap_tolerance 5 --interspersed_duplications_as_insertions --tandem_duplications_as_insertions --read_names --max_sv_size 1000000', restricting output SVs to <1 Mb.

## SVision

We applied SVision (v.1.3.8) to detect SV based on a pre-trained deep learning model (svision-cnn-model.ckpt). Parameters included a minimum read support threshold of 5, enabled graph output mode, and query name output with '-s 5 --graph –qname'.

## cuteSV

We ran cuteSV (v2.0.1) based on aligned HiFi reads with parameters '--max_cluster_bias_INS 1000 --diff_ratio_merging_INS 0.9 --max_cluster_bias_DEL 1000 --diff_ratio_merging_DEL 0.5 --min_size 50 --min_support 3 –genotype'.

## pbsv

We used the *discovery* and *calling* functions in pbsv (v2.8.0) to detect SVs, with a minimum SV length threshold set to 50 bp.

## Sniffles

We applied Sniffles (v2.0.7) on aligned sequences tagged by SAMtools' *calmd* tool, which reduces the base quality scores for mismatched bases. Sniffles parameters included '--max_cluster_bias_INS 1000 --diff_ratio_merging_INS 0.9 --max_cluster_bias_DEL 1000 --diff_ratio_merging_DEL 0.5 --min_size 50 --min_support 3 –genotype'.

## SNV calling

**HiFi data.** We employed a deep learning-based approach, Clair3 (v1.0.5), for SNP and INDEL calling on the 20 Holstein HiFi reads. We ran Clair3 using BAM files aligned by Minimap2 (v2.24) under the HiFi (hifi_sequel2) model specifically optimized for PacBio HiFi data.

## WGS data

We preprocessed short-read data using fastp v0.23.4 with default options. High-quality reads were aligned to the ARS-UCD v2.0 reference genome with bwa-mem2 (v2.1.1), followed by coordinate sorting with SAMtools. To reduce artifacts from polymerase chain reaction (PCR) and optical duplication, duplicate reads were marked and removed with GATK's MarkDuplicates (v4.1.4.1).

For variant calling, we applied DeepVariant (v1.6.1), a deep learning-based tool using a convolutional neural network, on the 20 Holstein samples, 8 samples, and 173 CDCB samples from Cooperative Dairy DNA Repository (CDDR)[68,69]. We then performed joint variant calling and merging across these datasets using GLnexus (v1.4.1).

## Hybrid HiFi and WGS data

To leverage both PacBio HiFi and Illumina WGS data for the 20 Holstein samples, we employed DeepVariant's hybrid model with '--model_type=HYBRID_PACBIO_ILLUMINA' on reads aligned by Minimap2 and bwa-mem2, respectively. The resulting calls were merged and jointly genotyped with GLnexus.

## SV calling and genotyping for WGS data

We utilized an integrated pipeline of Lumpy (v 0.2.13), SVTyper (v0.7.1), and CNVnator (v0.4.1) for per-sample SV calling[67]. Lumpy integrating both read-pair (RP) and split-read (SR) strategies, is a probabilistic framework that runs individually for each sample, and gives 4 main types of SV, including deletions (DEL), duplications (DUP), inversions (INV), and break ends (BND). To decrease run time and false positives, we utilized smoove (v0.2.8, https://github.com/brentp/smoove) to process alignment information. After individual calling by smoove call, we generated a merged set of variants across all samples using SVtools (v0.3.2) with *lsort* and *lmerge* functions. We then genotyped all variants for each sample using SVtools genotype and annotated with copy number (CN) information from a read depth (RD) strategy (CNVnator) by *SVtools copynumber*. Finally, *SVtools vcfpaste* and *prune* were used to merge and filter the identified SVs with default parameters, respectively.

To analyze genome dosage changes, we retained only DELs and DUPs as CNVs, excluding events shorter than 50 bp or longer than 1 Mb. We then filtered each CNV type using the optimal parameters, as recommended by the previous publication[66]. DELs were kept if MSQ > 100 and small DELs ≤1,000 bp were kept only if they had split-read support in at least one sample.

## Variations genotyping for WGS data using pangenome references

Using the H20D pangenome graph VCF generated with Minigraph-Cactus, we genotyped SVs from WGS data by PanGenie (v3.0.1), a short-read genotyper designed for detecting various genetic variants within a pangenome graph. PanGenie calculates genotypes by using read k-mer counts and a reference panel of known, fully assembled haplotypes. To refine the VCF, we filtered out the variations in the multi-allele graph VCF file with the following thresholds: sites within nested regions, variants over 100 kb, sites with >20% missing data, and those on sex or unplaced chromosomes. We decomposed bubbles to identify all nested variant alleles using a script (https://github.com/eblerjana/genotyping-pipelines/tree/main/prepare-vcf-MC). We then designated variations with sizes larger than 50 bp as PanGenie SV, while others were classified as SNV (SNP, MNP, and INDEL).

## Genome region intersection analyses

We used the intersect function in BEDTools (v2.31.1)[76] to identify overlapping genomic regions between datasets. For comparisons between the Holstein and M13H pangenomes, non-redundant regions unique to or shared between the 2 pangenomes were identified using BEDTools merge. To count SVs shared or unique to different SV calling strategies, we set a minimum overlap fraction threshold of 90% for both files using 'bedtools intersect -f 0.90 -r'.

## SV comparison

Truvari (v4.3.1)[77], an SV comparison, annotation, and analysis toolkit, was employed to assess precision, recall, F1 score, and genotype concordance across pangenome-based SV calling and assembly, mapping, and short-read methods for each individual. We used the bench function in Truvari with parameters '--dup-to-ins -r 2000 -C 2000 --no-ref a --passonly --sizemax 1000000 --sizemin 50', to restrict the length to between 50 bp and 1 Mb.

## SNV comparison

We compared the SNV call sets generated from 20 Holstein samples from HiFi data by Clair3[78], WGS by DeepVariant[79], and a hybrid HiFi/WGS model. The precision, sensitivity, and F-measure (similar to Truvari metrics) were calculated by the RTG Tools (v3.12.1) *vcfeval* function. All VCF files were summarized by *bcftools* (v 1.20) stats. To specify which genome region was included in the evaluation, the '-e' option was set for each gene/repeat annotation. The gene annotation file for

ARS_UCD2.0 was downloaded from NCBI. And the repeat regions were reported by RepeatMasker (v4.1.0)[80].

## SV annotation

We annotated repeat- and gene-overlapping SVs using the *findOverlaps* function in the GenomicRanges R package (v1.40.0), with a minimum overlap threshold of 1 bp. As previously described[67,81,82], the enrichment of specific genomic regions within SVs was assessed using fold tests with the whole genome as the background and Chi-squared tests for statistical significance. P-values were adjusted with the Bonferroni correction. Gene ontology (GO) and Kyoto Encyclopedia of Genes and Genomes (KEGG) pathway enrichment analyses were performed on gene lists of interest using the *enrichGO* and *enrichKEGG* functions from the clusterProfiler R package (v3.16.1, R v4.0.2), with a Bonferroni-adjusted significance threshold of 0.01.

## Phenotypes, dPTA, and correlation analysis

Individual trait data for 173 registered Holstein bulls were retrieved from the December 2022 genomic evaluations of the U.S. Council of Dairy Cattle Breeding (CDCB), which have been reported before[68,69]. These 46 phenotypes included predicted transmitting ability (PTA) values for: body depth (BD); body weight composite (BWC); calving trait composite (CT); cow conception rate (CCR); cow livability (LIV); dairy form (DF); daughter pregnancy rate (DPR); daughter calving ease (CDE); daughter stillbirth (DSB); displaced abomasum (DAB); early first calving (EFC); fat yield (FAT); feet and leg composite (FLC); final score (FS); foot angle (FA); fore udder attachment (FUA); front teat placement (FTP); gestation length (GL); health trait composite (HTH); heifer conception rate (HCR); heifer livability (HLV); ketosis (KET); mastitis (MAS); metritis (MET); milk fever/hypocalcemia (MFV); milk yield (MLK); productive life (PL); protein yield (PRO); rear leges rear view (RLR); rear legs side view (RLS); rear teat placement (RTP); rear udder height (RUH); rear udder width (RUW); residual feed intake (RFI); retained placenta (RPL); rump angle (RA); rump width (RW); sire calving ease (SCE); sire stillbirth (SSB); somatic cell score (SCS); stature (ST); strength (STRE); teat length (TL); udder cleft (UC); udder composite (UDC); and udder depth (UD). De-regressed PTA (dPTA) values for each trait were calculated as dPTA = PTA / reliability, as previously described[83].

## Genome-wide association study between SV/SNP and phenotypes

Using the SV/SNP genotype matrix from PanGenie, we removed rare SVs (MAF < 0.05). GWAS was conducted for each trait with EMMAX (v20120210) using a mixed linear model. The genetic relationship matrix used as input for EMMAX was based on 15,122 randomly selected SNPs in BTA1, spaced 10 kb apart, generated by PLINK (v1.90 beta) with --bp-space 10000. SNPs were called by GATK (v4.3.0.0) with HaplotypeCaller -ERC GVCF, CombineGVCFs, GenotypeGVCFs, and VariantFiltration using the filtering expression QD < 2.0 FS > 30.0 SQR > 3 MQ < 40.00 QUAL < 40.00, and SelectVariants -select-type SNP. The significance test was performed using randomly selected SVs and SNPs in equal numbers for the fat GWAS analysis.

## Ethics

Ethical permission to collect blood samples from cattle was approved by the U.S. Department of Agriculture, Agricultural Research Service, Beltsville Agricultural Research Center's Institutional Animal Care and Use Committee (Protocol 18-005).

## Reporting summary

Further information on research design is available in the Nature Portfolio Reporting Summary linked to this article.

## Data availability

Raw sequencing data used in this study, including HiFi long-read for 20 Holstein samples and 10 Jersey, Illumina whole-genome sequencing for 30 samples, and the 2 Hi-C data for assemblies are available from the NCBI BioProject database under accession numbers PRJNA1113979, PRJNA1129520 [https://www.ncbi.nlm.nih.gov/bioproject/ PRJNA1129520], PRJNA1224411 [https://www.ncbi.nlm.nih.gov/bioproject/ PRJNA1224411], and PRJNA1223899 [https://www.ncbi.nlm.nih.gov/bioproject/ PRJNA1223899], respectively. All datasets are publicly accessible without restrictions. Pangenome graph files (GFA format) and corresponding biallelic SV VCF annotations generated in this study have been deposited in Zenodo [https://doi.org/10.5281/zenodo.15313439]. All additional data supporting the findings of this study are provided in the paper and its Supplementary Information. Source data for this article are available on Zenodo [https://zenodo.org/records/15313439/files/Source%20Data.zip].

## Code availability

Additional scripts and resources are available on GitHub [https://github.com/xyxss/cattleHolPanSV][84]. To ensure permanent accessibility, an archived version of this repository has been deposited in Zenodo [https://doi.org/10.5281/zenodo.15313439].

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

## Acknowledgements

We thank Reuben Anderson, Donald Carbaugh, Christina Clover, Cecelia Niland, and Sara McQueeney for technical assistance and sample collection. We thank the Cooperative Dairy DNA Repository (CDDR) for providing access to sequence data that was used for this research project. We thank the Council on Dairy Cattle Breeding (Bowie, MD, USA) for genotype, phenotype, and pedigree data, Interbull (Uppsala, Sweden) for global trait evaluations, and the anonymous reviewers for many helpful comments. Mention of trade names or commercial products in this article is solely for the purpose of providing specific information and does not imply recommendation or endorsement by the US Department

of Agriculture (USDA). The USDA is an equal opportunity provider and employer.

This work was supported by AFRI grant numbers 2019-67015-29321, 2021-67015-33409, and 2024-67015-42295 from the USDA National Institute of Food and Agriculture (NIFA) Animal Genome Programs to G.E.L. and L.M. L.Y. is supported in part by the Young Scientists Fund of the National Natural Science Foundation of China (Grant No. 32302699).

## Author contributions

G.E.L. and T.P.L.S. conceived and supervised the study. L.Y., Y.G., and N.B. performed the computational and statistical analyses. L.Y. and G.E.L. drafted the manuscript. W.L., G.Z., C.L., and R.L.B.V.I. collected cattle samples. K.K. isolated high-molecular-weight DNA, prepared sequencing libraries, and performed HiFi sequencing and initial data processing. L.F. and J.B.C. collected and curated genomics datasets. C.P.V.T., B.D.R., and L.M. contributed to data interpretation and discussion of results. All authors reviewed and approved the final manuscript.

## Competing interests

The authors declare no competing interests.
