## [Transparent Peer Review file · Nature Communications]

Phased-assembly-driven pangenome graphs for structural variants genotyping and complex trait mapping in dairy cattle

Corresponding Author: Dr George Liu

Version 0:

Reviewer comments:

Reviewer #1

(Remarks to the Author)

This manuscript presents a very detailed analysis of structural variation in dairy cattle breeds. The authors have generated and repurposed sequencing data from several Holstein, Jersey, and other cattle breeds to build pangenome data structures using the commonly used Minigraph-Cactus toolkit. Aside from the pangenome graph approach, the authors also generated assembly-comparison-based, long-read mapping-based, and short-read mapping-based structural variation calls, along with SNP callsets. The authors compared different approaches and concluded that pangenome-based analysis improves the results.

I am not an expert on cattle genomes, and I have no opinions on the biological findings. Therefore, I focused rather on the methodology. All tools and approaches used by the authors are "industry standard", so no complaints there. I applaud the authors for a very extensive analysis; I myself would skip using some of the tools like SVision, pbsv, and CNVnator (nothing wrong with them); but maybe include dipcall for the assembly-based SV detection.

Reviewer #2

(Remarks to the Author)

I find the manuscript difficult to read. There are many results, graphs and numbers, as if one were reading supplementary notes, but considerations and explanations are scattered throughout the paragraphs. The text could be shortened to highlight the essential results more clearly. Furthermore, given the large amount of analysis, a GitHub repository reporting all scripts to reproduce them should be provided.

Authors should discuss the exclusion of other pangenome graph construction pipelines like the Pangenome Graph Builder and Minigraph. The latter is optimized to work with structural variants.

Line 45: I don't understand the meaning of the word "optimized" here. Optimized for which metric?

Line 61-63: "This limitation has driven the development of pangenomes—graph-based representations that incorporate multiple genomes to better capture sequence diversity." Pangenomes are not pangenome graphs. Pangenomes are sets of sequences, while pangenome graphs are one of the possible representations to represent pangenomes. The word pangenome is wrongly used as a synonym of pangenome graphs in several parts of the manuscript.

Line 69-70: "In this study, using the Minigraph-Cactus pipeline, we constructed and compared breed-specific (H20D) with multiple other pangenomes in detail." Are the words "pangenome graphs" missing after "breed-specific"?

Line 70-72: "We also confirmed that H20D not only far exceeded short-read approaches, but also surpassed both assembly- and read-based long-read SV callers." In what? At this line, it is still not mentioned.

Line 107: "Variants with a genotyping rate below 10% were excluded." What does it mean?

Line 122: About the "H20D vs. M13H" comparison, how can the authors distinguish between real biological differences and technical differences due to different assembly quality due to batch effects (different data, software, etc... between different

previously published assemblies)?

I don't find the correspondence between the text referring to panels in Figure S9 and the Supplementary Figure 9 I've downloaded, which just have a and b letters. There might be problems with the uploaded files.

Reviewer #3

(Remarks to the Author)

The authors constructed a breed-specific pangenome of Holstein cattle and used the pangenome to call variants in the assemblies used to construct the pangenome. They compared this pangenome-based calling pipeline to existing linear genome-based calling and genotyping methods and performed GWAS on the SVs in the pangenome. Although I find their results to be interesting, particularly the GWAS results, I believe that they are overstated. In particular, it is difficult to substantiate claims of superiority over other methods when there is no truth set.

Major comments

1. I find the title to be misleading as there was no real optimization of pangenome graph construction presented, only comparisons between mixed breed and haploid vs diploid pangenomes. Additionally, the mixed breed pangenome was not included in most of the analyses, contrary to the claim in page 2 line 37.
2. The authors spend a significant amount of space showing that the phased diploid pangenome produces better results than the haploid pangenome. Although it is useful to compare the diploid vs haploid pangenome and haploid pangenome vs haploid assembly-based calling methods, it is very unsurprising that the diploid methods perform better and it is worth mentioning only briefly. For example, the section "Improved SV detection and genotyping in phased diploid pangenomes compared to haploid pangenomes", and the corresponding Figure 3, could be combined with the previous section. Similarly, the section "Single-breed pangenome with the diploid model enhances SV genotyping and phasing in difficult genomic regions", while interesting and useful as an illustration of variants in a pangenome, only really shows the benefit of a diploid pangenome over a haploid one.
3. In comparisons between calling and genotyping methods, I found it difficult to assess the performance of the pangenome-based methods based only on comparisons to each other method individually, and I am not convinced by claims of superiority over other methods. For example, in the abstract (page 2 line 40, also repeated on page 3 line 71), the claim that the pangenome method "surpassed" and "exceeded" other methods and identified 10,000 more SVs per sample is unsubstantiated, and without a truth set it is difficult to determine if the new SVs are real variants or false positives. I would find it more convincing to do something like count the number of tools that support each variant found by any method, separating the ones that the MC method/PanGenie finds or misses.
4. The authors use pangenomes as a method of detecting variants in samples used to construct the pangenome. I would also like to see comparisons to pangenome methods that use the pangenome as a reference for new samples, as I expect that this is the more common use case. For example, using PanGenie on a graph with a held out sample or on the M13H pangenome, or the Giraffe-DeepVariant pipeline with different graph references for calling small variants.
5. Section "Pangenome improves SV genotyping in short reads": In this section, variant calls from H20D are compared to genotypes from PanGenie using H20D and reads from the samples in H20D. Both are then compared to short-read SV genotyping tools. I was unsure of which variants were genotyped by the short read tools, whether the tools themselves did both calling and genotyping, or if another call set from the pangenome or long reads was used. It seems to be an unfair comparison between the pangenome methods, which used assemblies and short reads from the same sample, and the non-pangenome methods, which only used short reads. The analysis with the Jersey pangenome is more fair but, as mentioned earlier, this would be a good place to use the mixed-breed pangenome or a pangenome with a held-out sample. Additionally, the calls from the H20D graph seem to be implicitly taken as a truth set; this should be explained explicitly.
6. Section "SNV consistency evaluation with the pangenome strategy": I found it a little hard to follow all the comparisons made here, but the hybrid DeepVariant model seems to be taken as the truth set without justification. I am not sure if it is justified.
7. Page 11 line 414: I am not convinced that simply calling more SNPs is an indication of "superior performance", as it is not shown whether these SNPs are real or errors.
8. I was confused by the different terms used to describe the pangenome construction and calling method. The authors use the abbreviation "MC" both for the assembly pipeline alone (page 4 line 79) and for their assembly and variant calling pipeline (page 5 line 159), and "H20D" is also used to refer to the variant calling pipeline (page 2 line 39, page 3 line 71, page 9 line 351), as well as the graph. It is also confusing to use the term "Minigraph-Cactus pipeline" to describe the method of constructing a graph and calling variants with vg deconstruct, as "Minigraph-Cactus" is usually used to refer only to the graph construction pipeline. It would be helpful to use consistent terms and to specify that the "MC" pipeline refers to both graph construction and deconstructing into a VCF. This should be done at least in the introduction, and the calling method should be briefly described the first time it is introduced.
9. Page 7 lines 247-254: I was confused by what methods and call counts were being compared in this paragraph. It seems

to me that the sentences “Compared to H20D, the shared proportion of SVs increased on average by 5.10% in H20H” and “the share proportion for the 9 strategies collectively decreased by 14.08%” directly contradict each other.

10. Page 10 line 394: What is the “MC on HiFi data” method? And which graph was used?

11. It would be useful to include a runtime and memory-use comparison between the MC method and the other methods.

Minor comments

12. Section “SV genotyping for WGS data using a pangenome reference enhances detection accuracy”: This section seems to be mis-named, or the section title added by accident.

13. Section “Pangenome SV-based GWAS in 173 WGS samples”: How were complex variants dealt with for the GWAS analysis? Based on the supplement it seems that GWAS was only done using SNPs, not SVs.

14. Page 4 line 110: How are nested variants and multi-allelic variants represented? What is the definition of a COMPLEX variant?

15. Page 4 line 113: It is odd that there are so many more insertions than deletions. I’m curious if the authors have any theories.

16. Page 9 line 335: How were the ancestral alleles determined?

17. Page 9 line 353: It was unclear what was being compared here- based on the figure, it seems to be the percent of variants called on H20D using their MC method that were also called by PanGenie, and vice versa?

18. Page 9 line 353: A better pangenome reference improves SV genotyping accuracy as compared to what? Only one pangenome reference was used.

19. Page 10 line 382: When comparing PanGenie on the J10D graph with the calls from H20D, the calls in common were reported as an overall count; could this be reported as a percent as well, as it is for the other methods?

20. Page 12 line 466: The SNP-based GWAS was not described.

21. Page 13 line 483: I’m not sure if the SVs can be classified as “breed-specific” if they were not checked for in other breeds.

22. Page 3 line 70: “we constructed and compared breed-specific (H20D) with multiple other... -> “a breed-specific pangenome (H20D)”

23. Page 7 line 239: “An average of 16.24k SVs was identified” -> “were identified”

24. Page 8: Many references to Figure 4 are missing or incorrectly reference Figures 3 or S9.

25. Page 8 line 296/Figure 4a: The text says that Sniffles detected two deletions but in the figure there appears to only be one. Similarly, the insertion seems to be two insertions in the graph.

26. Figure 4c: I found the variants floating in the middle of the graph to be confusing. Could the paths on the graph be highlighted a different way? Maybe an image from the sequence tube map could be more helpful?

27. Figure 4d: I don’t think the orientation of the nodes matters, since there don’t appear to be any inversions. The “30bp INS” and “32bp INS” arrows also seem to be switched.

28. Page 8 line 304: “Unlike other pangenomes” but only the haploid pangenome was compared.

29. Page 9 line 310/Figure 4c: The path of the insertion in the text doesn’t match the path in the graph

30. Page 10 line 396: PG(JerRef) is not named consistently with the other methods

31. Page 10: I think this is missing/mixing up references to Figure 6.

32. Page 11 line 431: “multiple SV” -> “multiple SVs”

33. Discussion: The first sentence of the discussion is repeated.

34. Figure 1: I was confused by panels a and c. They appear to be showing the same data, but with different results: the shared length goes down to ~2.7G in panel a and ~1.8G in panel c.

35. Figure 2: Panel a is missing an x-axis label

36. Missing citations for Clair3 and DeepVariant

37. Page 17 line 657: The samples from CDDR were never mentioned in the paper

Reviewer #4

(Remarks to the Author)

In this study, Yang et al. contribute to the growing body of work advocating for the use of pangenomes to genotype complex structural variants (SVs), which remain relatively understudied. The manuscript appropriately emphasizes the need to understand the genetic sourcing behind pangenome builds to effectively assess complex SVs. While the authors consistently apply previously published methods, the frequent switching between phased, unphased, and mixed-breed pangenome builds complicates interpretation and makes it difficult to extract clear conclusions. The extensive number of analyses reflected in 13 and 30 supplemental figures and tables, respectively, that demonstrates a commendable thoroughness in method optimization. However, this also becomes a liability, as many comparisons yield overlapping results (e.g., more diverse pangenomes identify more SVs overall but may struggle with complex SVs), which obscures the key takeaways.

The manuscript would benefit from simplification, particularly through the removal or relegation of non-essential analyses. For example, the sections on SNV calling using pangenomes are less impactful, given that this is not a major advantage of pangenome references compared to linear T2T-like genomes. A concise summary of the SNV results in the main text, with detailed results moved to the supplement, would improve focus. Additionally, some of the conclusions on pangenome builds and SV genotyping reiterate findings already reported, such as those in Pausch et al., *Genome Research* (2024), which limits the novelty of the work. That said, this study does offer valuable insights, particularly regarding the differential performance of SV genotyping using high- versus low-diversity pangenome graphs.

Other minor points by section:

Introduction

The introduction seems to lack a clear narrative to effectively set up the main theme of optimizing pangenome graph construction. I recommend a partial rewrite that more concisely frames the study within the context of prior work. This should include a summary of relevant studies in cattle particularly those from Pausch et al. as well as the key human pangenome efforts that define the current state of the art. Emphasis should be placed on short-read genotyping performance using pangenomes, drawing from foundational work by Garrison et al. and Paten et al. This would provide a stronger conceptual foundation and better contextualize the goals and significance of the current study.

p. 3 line 56- SVs have been shown to (need citations here).

Results

p. 4. Line 109. I think this is the first use of the term COMPLEX as a classification for some SV types. This needs to be defined at some point.

p. 5 line 163-64. Briefly explain why SVs on ChrX from female samples were retained for this analysis.

p. 6. Lines 87-191. After this conclusion of haploid limitations don't the authors focus on diploid modelling for the remainder of the study.

p. 7. Lines 225-26. This section can be condensed by deemphasizing long read mapping consistency as long-read mapping inferiority is already established earlier.

p. 9. Lines 332-345. This section can be moved to supplement and is an example of my recommendation to move all the SNV characterizations to supplement.

p. 9. Line 346. This section is the most important narrative to expand in this reviewers opinion even at the expense of other sections. Cohorts with vast numbers of short-read samples are available for genotyping and GWAS study. The limitation in their use for SV assessment needs to be better understood across species with different evolved structures, especially those domesticated.

p. 10. Line 390. Same argument as before for the movement of these analyses to the supplement.

p. 11. Line 421. Use SV abbreviation here and note which genotyping method was used. I assume PanGenie. Why were these 173 Holstein's chosen?

p. 11. Lines 436-37. How so on their influence? Detection alone or similar agreements for SNVs in other GWAS studies.

p. 11. Line 449. What does SV-related genes mean? Some clarification would be better.

p. 12. Lines 463-65. This statement in the results could perhaps be moved to discussion. Same comment for lines 471-75.

Discussion.

The discussion revisits themes explored in earlier studies on the use of pangenomes for genotyping, with the primary

novelty here being the specific application to cattle genomes. However, the discussion lacks citations that contrast these findings with prior work, which limits the reader's ability to contextualize the results. It is unclear why the authors chose not to reference earlier studies, and doing so would strengthen the discussion by clarifying how this work builds upon or diverges from established findings.

p. 13. I don't think the term practical applies here. It is currently not practical to genotype thousands of Holsteins that have been sequenced at lower coverage, to address cost concerns, for cost and imputed for SNVs.

p. 13. Lines 514-15. I think a statement that includes "offers improved accuracy" over existing genome selection indices is premature until further studies prove this hypothesis. It certainly could once models incorporate SVs for larger numbers of cattle.

Methods.

p. 15. Line 553. Gao et al. under review needs clarification if this study will overlap significantly.

p. 15. Line 562. Cite the study that refers to previously analyzed cattle genomes.

p. 18. Line 717. What is this reference?

p. 19. Lines 750-51. This sentence is confusing. Clarify the methods.

I commend the authors for generating high-quality data using appropriate techniques, which have been carefully analyzed, interpreted, and presented in detail. I did not identify any major methodological flaws. However, the study would benefit from clearer messaging and more explicit acknowledgments that many of the findings align with and confirm earlier work on pangenome genotyping in other species. The primary significance of this work lies in providing additional validation that can inform the computational design of future studies particularly those aiming to enhance the efficiency of SV genotyping across large whole-genome sequencing cohorts.

Version 1:

Reviewer comments:

Reviewer #2

(Remarks to the Author)

The authors addressed all my comments. I appreciate the GitHub repository, but it still needs some tweaking. For example, the "annotate_vcf.py" file is missing.

Reviewer #3

(Remarks to the Author)

I thank the authors for the significant changes that they have made to the paper and for their thorough responses to my comments. I find this draft to be much more clearly written and the results presented in a more honest and impactful way. I have only a few remaining concerns.

1. Response to Q21: The majority of this paper is spent comparing methods, and resource consumption is an important consideration when choosing a method. I would still like to see an analysis of runtime and memory, even if it is only a rough estimate of only the MC pipeline mentioned briefly.

2. Page 10 line 395: In their response to Q16, the authors say that they did not designate a truth set. While they did show pairwise analyses in the figure and did not specifically designate a truth set, in the text they present F1, precision, and sensitivity values relative to the HB method, which I find to be implicitly making a claim about correctness of other methods based on the HB results. Although this is not an incorrect result, I find it to be very misleading unless it is clearly and explicitly described why it was done.

3. Response to Q13. I believe the authors misinterpreted the analysis I suggested; the analyses they referenced in their response still perform pairwise comparisons between methods. I was suggesting pooling all variants from all methods and stratifying them by how many tools support each variant. Then compare each method of interest to the stratified set of pooled variants. However, while I still believe that this analysis would be relatively easy and informative, the authors have already mostly addressed my main concern that their analyses didn't support their claims by softening their claims so this is less important.

4. Response to Q15. I agree with the authors that the variants from the H20D graph are a reasonable truth set here, but this should be explained explicitly in the text.

5. Page 3 line 75: "Besides enhancing the efficiency and accuracy of genome assembly, variation-aware graphs..." this needs a citation. I am not aware of any such work.

6. Page 3 line 80: I would consider the graph itself to be the catalog. Also Giraffe is a mapper, vg call is the genotyper that is sometimes used with Giraffe.
7. Page 5 line 150: This could reference the methods for the definition of COMPLEX variants, and also for how multi-allelic variants are converted to bi-allelic.
8. Page 6 line 209: There is an extra parentheses after "haploid assemblies"
9. Page 9 line 317: Should this figure reference be 4a?
10. Page 9 line 346: Is H20D referring to the MC pipeline with H20D? Additionally, these are both pangenome methods so this does not yet "underscore the value of a pangenome reference".
11. Figures 2d, 2f, 6d, and 6f: What do the sizes of the squares/circles mean?

Reviewer #4

(Remarks to the Author)

In reviewing the authors' responses and how they addressed prior critiques, I find that the manuscript now presents a much-improved interpretation of pangenome applications and relevance. Some sections could still benefit from more concise writing, and a few results remain somewhat overstated in their novelty. Nonetheless, the manuscript provides a solid and informative account of how pangenomes can be effectively utilized.

**REVIEWER COMMENTS**

Reviewer #1 (Remarks to the Author):

This manuscript presents a very detailed analysis of structural variation in dairy cattle breeds.
The authors have generated and repurposed sequencing data from several Holstein, Jersey, and
other cattle breeds to build pangenome data structures using the commonly used Minigraph-
Cactus toolkit. Aside from the pangenome graph approach, the authors also generated assembly-
comparison-based, long-read mapping-based, and short-read mapping-based structural variation
calls, along with SNP callsets. The authors compared different approaches and concluded that
pangenome-based analysis improves the results.

**Q1.** I am not an expert on cattle genomes, and I have no opinions on the biological findings.
Therefore, I focused rather on the methodology. All tools and approaches used by the authors are
"industry standard", so no complaints there. I applaud the authors for a very extensive analysis; I
myself would skip using some of the tools like SVision, pbsv, and CNVnator (nothing wrong
with them); but maybe include dipcall for the assembly-based SV detection.

**AU:** Thank you for your suggestion regarding *dipcall*. We acknowledge its strengths in
assembly-based small variant benchmarking and appreciate its contribution to the field. In this
study, our primary focus with long-read sequencing was to improve the detection of structural
variants from haplotype-resolved assemblies. For this purpose, we relied on tools such as PAV
and SVIM-asm, which are specifically optimized and widely adopted for identifying large
insertions, deletions, and complex SVs—the central focus of our analyses. Together, these
methods provided a robust and comprehensive SV callset. For small-variant calling, current
short-read sequencing approaches remain sufficiently robust. Incorporating *dipcall* would add
significant computational cost and extend beyond the scope and objectives of this work. That
said, we will consider its integration in future studies aimed at more comprehensive assembly-
based variant analyses.

Reviewer #2 (Remarks to the Author):

**Q2.** I find the manuscript difficult to read. There are many results, graphs and numbers, as if one
were reading supplementary notes, but considerations and explanations are scattered throughout
the paragraphs. The text could be shortened to highlight the essential results more clearly.
Furthermore, given the large amount of analysis, a GitHub repository reporting all scripts to
reproduce them should be provided.

**AU:** Thank you for this valuable feedback. We have carefully revised the manuscript to improve
clarity and readability by reorganizing the presentation to more tightly integrate explanations
with key results. Non-essential details have been condensed or moved to the Supplementary,
which helps streamline the main text and highlight the essential findings more clearly.
Additionally, we have provided a GitHub repository containing all scripts and resources
necessary to reproduce our analyses, accessible here: <https://github.com/xyxss/cattleHolPanSV>
(also referenced on L761). We hope these improvements make the manuscript easier to follow
and enhance transparency and reproducibility.

**Q3.** Authors should **discuss** the exclusion of other pangenome graph construction pipelines like
the Pangenome Graph Builder and Minigraph. The latter is optimized to work with structural
variants.

**AU:** Thank you for your insightful comment regarding the inclusion of other pangenome graph
construction pipelines such as Minigraph and the Pangenome Graph Builder (PGGB). We fully
acknowledge the value of these tools in structural variant detection. Minigraph is optimized for
the rapid construction of pangenome graphs, particularly excelling at representing structural
variants across multiple genomes with relatively low computational cost. However, Minigraph
alone is primarily a mapping-based approach and may have limitations in capturing complex
variations fully. PGGB, on the other hand, offers a comprehensive graph construction using a
progressive multiple sequence alignment approach, which can resolve complex structural
variants more accurately but at the expense of substantially higher computational resources and
longer runtimes. In our recent publication, we have tested Minigraph as a preliminary analysis
(<https://pubmed.ncbi.nlm.nih.gov/40258473/>). In this study, we employed the combined
Minigraph+Cactus (MC) approach, which integrates the speed and scalability of Minigraph with
the alignment accuracy of Cactus. This hybrid method enables a more precise and complete
representation of complex structural variants while maintaining reasonable computational
efficiency. Based on these considerations, we selected Minigraph+Cactus as the optimal pipeline
to balance performance, accuracy, and scalability for our large-scale cattle pangenome analysis.
We agree that PGGB and Minigraph have unique strengths, and future work could include
systematic comparisons across these tools to further improve structural variant detection.

**Q4.** Line 45: I don't understand the meaning of the word "optimized" here. Optimized for which
metric?

**AU:** By "optimized," we mean that using phased assemblies as input is key to constructing
pangenome graphs that accurately represent structural variants while maintaining computational
efficiency. We have revised the title to "Phased-assembly-driven pangenome graphs for
structural variation genotyping and complex trait mapping in dairy cattle".

**Q5.** Line 61-63: "This limitation has driven the development of pangenomes—graph-based
representations that incorporate multiple genomes to better capture sequence diversity."
Pangenomes are not pangenome graphs. Pangenomes are sets of sequences, while pangenome
graphs are one of the possible representations to represent pangenomes. The word pangenome is
wrongly used as a synonym of pangenome graphs in several parts of the manuscript.

**AU:** Thank you for highlighting this important distinction. We agree that "pangenomes" refer to
sets of sequences, while "pangenome graphs" are one way to represent them. We have carefully
revised the manuscript to avoid using "pangenome" as a synonym for "pangenome graph." In
Line 57 and throughout the text, we now specifically use "pangenome graphs" where appropriate
to ensure clarity and accuracy.

**Q6.** Line 69-70: "In this study, using the Minigraph-Cactus pipeline, we constructed and
compared breed-specific (H20D) with multiple other pangenomes in detail." Are the words
"pangenome graphs" missing after "breed-specific"?

**AU:** Yes, "pangenome graphs" was intended in that sentence. We have revised it for clarity to
read: "In this study, using the Minigraph-Cactus pipeline (Hickey et al. 2024), we constructed

and compared breed-specific pangenome graphs (H20D) with multiple other pangenome graphs
in detail.”

**Q7.** Line 70-72: “We also confirmed that H20D not only far exceeded short-read approaches, but
also surpassed both assembly- and read-based long-read SV callers.” In what? At this line, it is
still not mentioned.

**AU:** We have revised the text to clarify that H20D not only significantly exceeded short-read
approaches but also outperformed assembly- and read-based long-read SV callers in terms of
structural variant detection and comprehensiveness. Specifically, as detailed in Line 38, “It also
significantly improved SV detection and genotyping relative to graphs built across breeds or
from fewer/unphased assemblies, with particular advantages in complex regions.”

**Q8.** Line 107: “Variants with a genotyping rate below 10% were excluded.” What does it mean?

**AU:** Variants with a genotyping rate below 10% were excluded, meaning only variants
successfully genotyped in at least 10% of samples were retained. We have revised the sentence
to: “Variants with a genotyping rate above 10% were kept.”

**Q9.** Line 122: About the “H20D vs. M13H” comparison, how can the authors distinguish
between real biological differences and technical differences due to different assembly quality
due to batch effects (different data, software, etc... between different previously published
assemblies)?

**AU:** We appreciate this important point regarding potential batch effects. To minimize technical
variation within our study, all 20 Holstein assemblies in the H20D set were generated using the
same pipeline (hifiasm) with consistent parameters and comparable input data quality. In
contrast, the M13H assemblies were publicly available and produced using various methods and
datasets, which may introduce technical differences. We therefore interpret the comparison
between H20D and M13H cautiously, focusing on broad trends rather than definitive biological
conclusions. On L201: we have added “It is noted that this section focuses on broad trends rather
than definitive biological conclusions, as it is not possible to clearly separate true biological
differences from technical variation caused by differences in assembly quality or batch effects.”

**Q10.** I don’t find the correspondence between the text referring to panels in Figure S9 and the
Supplementary Figure 9 I’ve downloaded, which just have a and b letters. There might be
problems with the uploaded files.

**AU:** In the main text, we have replaced occurrences of “Figure S9” with “Figure 4” to ensure
consistency with the figure numbering. The updated figure file has been uploaded accordingly,
and figure numbering has been adjusted throughout the manuscript for clarity and consistency.

Reviewer #3 (Remarks to the Author):

The authors constructed a breed-specific pangenome of Holstein cattle and used the pangenome
to call variants in the assemblies used to construct the pangenome. They compared this
pangenome-based calling pipeline to existing linear genome-based calling and genotyping
methods and performed GWAS on the SVs in the pangenome. Although I find their results to be
interesting, particularly the GWAS results, I believe that they are **overstated**. In particular, it is
difficult to substantiate claims of superiority over other methods when there is no **truth set**.

Major comments

Q11. 1. I find the title to be misleading as there was no real optimization of pangenome graph construction presented, only comparisons between mixed breed and haploid vs diploid pangenomes. Additionally, the mixed breed pangenome was not included in most of the analyses, contrary to the claim in page 2 line 37.

AU: See our answer to Q4. We agree that the original title could be misleading and have revised it to “Phased-assembly-driven pangenome graphs for structural variation genotyping and complex trait mapping in dairy cattle.” By the “*mixed breed*” pangenomes, we mean pangenomes constructed from assemblies from multiple pure cattle breeds (beyond Holstein) as opposed to those built entirely within a single breed (Holstein). No hybrid breeds were included. We changed “mixed” to “multiple” on L57.

Q12. 2. The authors spend a significant amount of space showing that the phased diploid pangenome produces better results than the haploid pangenome. Although it is useful to compare the diploid vs haploid pangenome and haploid pangenome vs haploid assembly-based calling methods, it is very unsurprising that the diploid methods perform better and it is worth mentioning only briefly. For example, the section “Improved SV detection and genotyping in phased diploid pangenomes compared to haploid pangenomes”, and the corresponding Figure 3, could be combined with the previous section. Similarly, the section “Single-breed pangenome with the diploid model enhances SV genotyping and phasing in difficult genomic regions”, while interesting and useful as an illustration of variants in a pangenome, only really shows the benefit of a diploid pangenome over a haploid one.

AU: We agree that the advantage of diploid over haploid pangenomes is expected. Accordingly, we have condensed the results for this comparison. The “Improved SV detection... L275” section and the “Single-breed pangenome... L305” section have been cut down 50%, combining them into single-paragraph summaries. Detailed analyses have been moved to the Supplementary Notes. This preserves the key findings for completeness while avoiding overemphasis on an anticipated result.

Q13. 3. In comparisons between calling and genotyping methods, I found it difficult to assess the performance of the pangenome-based methods based only on comparisons to each other method individually, and I am not convinced by claims of superiority over other methods. For example, in the abstract (page 2 line 40, also repeated on page 3 line 71), the claim that the pangenome method “surpassed” and “exceeded” other methods and identified 10,000 more SVs per sample is unsubstantiated, and without a truth set it is difficult to determine if the new SVs are real variants or false positives.

I would find it more convincing to do something like count the number of tools that support each variant found by any method, separating the ones that the MC method/PanGenie finds or misses.

AU: We acknowledge the limitation of not having a complete truth set to fully validate new SV calls. Accordingly, we revised the abstract and main text to avoid unqualified claims of superiority. Rather than stating that H20D “surpassed” or “exceeded” other methods, we now describe it as providing “a more comprehensive SV profile” relative to short-read and other long-read SV callers. For example, as noted in the main text (Line 82), H20D recovered 34.35–39.63% unique SVs not detected by assembly- or read-based long-read methods.

We also agree that counting the number of methods supporting each variant would strengthen the
analysis. To address this, Table S14 and Figure 3d,e summarize the overlaps between MC-
derived SVs (from H20D or H20H) and those called by the other nine methods, providing a
relative measure of confidence. Please also see our answer to Q19 for details. For PanGenie SV
calls, similar comparisons are presented in Figure 5d and Table S22. For example, when
comparing MC SV calls (Tool1) from H20D with PanGenie SVs (Tool2), the shared rate was
83.54% with Tool1 normalization (Table S22, Cell I66) or 69.79% with Tool2 normalization
(Table S22, Cell K66).

**Q14.** 4. The authors use pangenomes as a method of detecting variants in samples used to
construct the pangenome. I would also like to see comparisons to pangenome methods that use
the pangenome as a reference for new samples, as I expect that this is the more common use
case. For example, using PanGenie on a graph with a held out sample or on the M13H
pangenome, or the Giraffe-DeepVariant pipeline with different graph references for calling small
variants.

**AU:** We agree that testing on new, unseen samples is an important and more representative use
case for pangenomes. For long-read genotyping, we have partially addressed this by testing
Holstein data on M13H, J10D, and J20H graphs. For short-read genotyping, our PanGenie
analysis already reflects this scenario, as it is designed to genotype new individuals from an
existing graph without reassembly. In this study, our primary goal was to evaluate the
performance of phased-diploid pangenomes under controlled conditions, which is why we used
the same samples for both graph construction and evaluation—ensuring a direct comparison
between graph configurations without confounding sample differences. We acknowledge that
this setup may overestimate accuracy compared to real-world applications. As you suggest, a
leave-one-out strategy or using external datasets (e.g., PanGenie on a held-out sample or the
Giraffe-DeepVariant pipeline on different graph references) would be a valuable next step.
However, due to the limited availability of suitable long-read datasets for dairy cattle, we could
not extend this evaluation for long-read-based genotyping in the current study. We have added
this as a limitation and direction for future work in the revised manuscript.

**Q15.** 5. Section “Pangenome improves SV genotyping in short reads”: In this section, variant
calls from H20D are compared to genotypes from PanGenie using H20D and reads from the
samples in H20D. Both are then compared to short-read SV genotyping tools. I was unsure of
which variants were genotyped by the short read tools, whether the tools themselves did both
calling and genotyping, or if another call set from the pangenome or long reads was used. It
seems to be an unfair comparison between the pangenome methods, which used assemblies and
short reads from the same sample, and the non-pangenome methods, which only used short
reads. The analysis with the Jersey pangenome is more fair but, as mentioned earlier, this would
be a good place to use the mixed-breed pangenome or a pangenome with a held-out sample.
Additionally, the calls from the H20D graph seem to be implicitly taken as a **truth set**; this should
be explained explicitly.

**AU:** In this section, short-read SV tools (e.g., Lumpy, CNVnator, SVtools) performed both
variant discovery and genotyping directly from short reads. For PanGenie, we used the H20D
graph to genotype the same samples’ short reads, without additional variant discovery. The
variants from the H20D graph—derived from high-quality phased assemblies—were treated as
the reference call set/true set for comparison, as they are expected to capture a more complete

and accurate SV landscape than short-read-based discovery. We acknowledge that this setup may
favor pangenome-based genotyping, since it leverages assemblies and short reads from the same
individuals, whereas short-read tools rely solely on read-based discovery. To address this, we
performed an additional, more balanced evaluation using 10 independent Jersey short-read
samples (~31.42×, labeled PanGenie-Jer) with the Holstein pangenome graph (H20D) as the
reference. These Jersey samples are completely separate from the Holstein dataset used to
construct H20D (see L382). On average, PanGenie-Jer detected 27.28k autosomal SVs, of which
11.53k (42.26%) were shared with the 9 genotyping strategies evaluated. This shared rate was
approximately twice that of Lumpy (3.73k, 17.00%), three times that of CNVnator (0.37k,
1.73%), and forty times that of SVtools (2.82k, 12.46%) (Table S24). These results indicate that
even when tested on an entirely different breed, pangenome-based genotyping retains a
substantial accuracy advantage over conventional short-read methods.

**Q16.** 6. Section “SNV consistency evaluation with the pangenome strategy”: I found it a little
hard to follow all the comparisons made here, but the hybrid DeepVariant model seems to be
taken as the truth set without justification. I am not sure if it is justified.

**AU:** We did not designate the hybrid DeepVariant model as a truth set. Instead, we performed
pairwise comparisons among all seven strategies without assuming any single method to be the
ground truth. We avoided designating a truth set because no high-confidence, fully validated
SNV dataset exists for these cattle populations, and each method may detect different subsets of
true variants. Our goal was therefore to assess agreement patterns across methods rather than
benchmark against a presumed gold standard.

**Q17.** 7. Page 11 line 414: I am not convinced that simply calling more SNPs is an indication of
“superior performance”, as it is not shown whether these SNPs are real or errors.

**AU:** We modified it to “indicating its greater sensitivity in challenging genomic contexts,
although accuracy was not directly assessed.”

**Q18.** 8. I was confused by the different terms used to describe the pangenome construction and
calling method. The authors use the abbreviation “MC” both for the assembly pipeline alone
(page 4 line 79) and for their assembly and variant calling pipeline (page 5 line 159), and
“H20D” is also used to refer to the variant calling pipeline (page 2 line 39, page 3 line 71, page 9
line 351), as well as the graph. It is also confusing to use the term “Minigraph-Cactus pipeline”
to describe the method of constructing a graph and calling variants with *vg deconstruct*, as
“Minigraph-Cactus” is usually used to refer only to the graph construction pipeline. It would be
helpful to use consistent terms and to specify that the “MC” pipeline refers to both graph
construction and deconstructing into a VCF. This should be done at least in the introduction, and
the calling method should be briefly described the first time it is introduced.

**AU:** We agree that our terminology for pangenome construction and variant calling was
inconsistent and could cause confusion. In the revised manuscript (L124 and L584), we now
explicitly define “MC pipeline” at first mention as the combination of Minigraph-Cactus graph
construction and variant extraction using *vg deconstruct*. We also consistently use “H20D” to
refer only to the Holstein pangenome graph, and not to the variant calling pipeline. Furthermore,
we have clarified in the Introduction and Methods (L1003) that when we refer to the MC
pipeline, we mean both graph construction and variant deconstruction into VCF format. This

ensures consistent usage throughout the manuscript and avoids ambiguity between the graph and
the variant calling workflow.

**Q19.** 9. Page 7 lines 247-254: I was confused by what methods and call counts were being
compared in this paragraph. It seems to me that the sentences “Compared to H20D, the shared
proportion of SVs increased on average by 5.10% in H20H” and “the share proportion for the 9
strategies collectively decreased by 14.08%” directly contradict each other.

**AU:** Sorry for the confusion. All numbers and their calculations (Cells and Formulas) are
provided in Table S14, which reports SV region shared rates between haplotype-resolved
(H20D) and primary haploid (H20H) pangenome graphs, as compared with the other nine
strategies (four assembly-based and five read-based SV callers).

When using Tool1 as the 100% reference, the average shared proportion of SVs across the other
nine strategies increased by 5.10% in H20H compared with H20D (Table S14, Cell I37), largely
driven by high concordance with svim-asm_haploid (19.12%, Cell I27) and pav_haploid
(19.06%, Cell I28). In contrast, when using Tool2 as the 100% reference, the average shared
proportion decreased by 14.08% in H20H relative to H20D (Table S14, Cell K37), primarily due
to the seven non-haploid methods (svim-asm_diploid, pav_diploid, cutesv, pbsv, sniffles, svim,
and svision), which showed decreases ranging from -13.59% to -19.56% (Cells K29-K35).
Together, these results indicate that H20H shows less stability and greater variability than H20D
when cross-referenced with other methods.

On L23 of Supplemental Notes, we modified to “We then assessed the consistency of SVs
detected in the H20D pangenome graph with those identified by nine long-read SV calling
strategies. As shown in Table S14, when using Tool1 as the 100% reference, the average shared
proportion of SVs across the other nine strategies increased by 5.10% in H20H compared with
H20D (Cell I37). This increase was largely driven by high concordance with the haploid
assembly-based strategies svim-asm_haploid (19.12%, Cell I27) and pav_haploid (19.06%, Cell
I28) (Figure 3d–e, Table S14), whereas the remaining methods showed only a negligible increase
of 1.10% (Cell I38). In contrast, when using Tool2 as the 100% reference, the average shared
proportion decreased by 14.08% in H20H relative to H20D (Cell K37). This reduction was
mainly attributable to the seven non-haploid methods (svim-asm_diploid, pav_diploid, cutesv,
pbsv, sniffles, svim, and svision), which showed decreases ranging from -13.59% to -19.56%
(Cells K29–K35), while the two haploid assembly-based strategies contributed minimally (-
0.94% and -0.15%, Cells K27 and K28).” On L39 of Supplemental Notes, we added “Together,
these results suggest that H20D is more stable and exhibits less variability than H20H when
cross-referenced with other methods.”

**Q20.** 10. Page 10 line 394: What is the “MC on HiFi data” method? And which graph was used?

**AU:** On L391, it was modified to “Minigraph-Cactus with *vg deconstruct* (MC(HiFi))”. Also on
L111, 206, 600, and 1074, we changed to “Minigraph-Cactus pipeline for pangenome graph
generating and *vg deconstruct* for the VCF exporting on HiFi data.”

**Q21.** 11. It would be useful to include a runtime and memory-use comparison between the MC
method and the other methods.

**AU:** Since this manuscript focuses primarily on biological findings rather than method
development, we did not prioritize runtime and memory-use benchmarking. Our goal was to

evaluate variant detection performance and biological insights rather than computational
optimization.

Minor comments

**Q22.** 12. Section “SV genotyping for WGS data using a pangenome reference enhances
detection accuracy”: This section seems to be mis-named, or the section title added by accident.

**AU:** We deleted “SV genotyping for WGS data using a pangenome reference enhances detection
accuracy”.

**Q23.** 13. Section “Pangenome SV-based GWAS in 173 WGS samples”: How were complex
variants dealt with for the GWAS analysis? Based on the supplement it seems that GWAS was
only done using SNPs, not SVs.

**AU:** The GWAS in this section were performed on either SV or SNP genotypes. See our answer
to Q24 about Complex SV genotyping.

**Q24.** 14. Page 4 line 110: How are nested variants and multi-allelic variants represented? What
is the definition of a COMPLEX variant?

**AU:** Nested variants and multi-allelic variants are represented using a graph-based approach.
Multi-allelic variants are highlighted in the provided figures with red boxes. To facilitate
analysis, these multi-allelic variants have been decomposed into bi-allelic variants, as illustrated
below the figures.

In the variant representation:

- • The **ID** field reflects the reference (REF) allele sequence as it traverses the graph nodes.
- • The **INFO** field contains information on the alternative (ALT) alleles, representing the
different paths through the graph.

Classification of variants was performed using PanGenie.

Regarding **COMPLEX variants**, these are defined as variants that differ from the REF allele by
at least two base pairs and are not classified simply as deletions (DEL) or insertions (INS). In
other words, COMPLEX variants include combinations or substitutions that cannot be captured
by straightforward single-base or indel changes.

Please see the two figures. The multi-allelic variants are highlighted with red boxes. These have
been converted into bi-allelic variants as shown below. The ID reflects the variation of the REF
allele as it traverses through the graph nodes, while the INFO field displays the different paths
for the ALT alleles. The classification was performed using PanGenie.

An example of pangenome VCF file

```
##INFO=<ID=AC,Number=A,Type=Integer,Description="Total number of alternate alleles in called genotypes">
##INFO=<ID=AF,Number=A,Type=Float,Description="Estimated allele frequency in the range (0,1]">
##INFO=<ID=NS,Number=1,Type=Integer,Description="Number of samples with data">
##INFO=<ID=AN,Number=1,Type=Integer,Description="Total number of alleles in called genotypes">
##INFO=<ID=LV,Number=1,Type=Integer,Description="Level in the snarl tree (0=top level)">
##INFO=<ID=PS,Number=1,Type=String,Description="ID of variant corresponding to parent snarl">
##INFO=<ID=AT,Number=R,Type=String,Description="Allele Traversal as path in graph">
```

#CHROM	POS	ID	REF	ALT	QUAL	FILTER	INFO	FORMAT
1	176493	>8370>8377	CTG	TAT,TAG,CAG	60	.	AC=2,1,45;AF=0.0333333,0.0166667,0.75;AN=60;AT=>8370>8371>8372>8375>8377>8370>8373>8374>8376>8377>8370>8373>8374>8375>8377>8370>8371>8374>8375>8377;NS=37;LV=0;ID=1-176493-COMPLEX->8370>8373>8374>8375>8376>8377-3,1-176493-COMPLEX->8370>8373>8374>8375-2,1-176494-SNV->8371>8374>8375-1	GT
1	176498	>8377>8380	CC	C,CA	60	.	AC=0,1;AF=0.0169492;AN=59;AT=>8377>8378>8380>8377>8380>8377>8379>8380;CONFLICT=GCA_905123885;NS=36;LV=0;ID=1-176498-DEL->8377>8380-1,1-176499-SNV->8377>8379>8380-1	GT
1	176501	>8380>8383	G	T	60	.	AC=1;AF=0.0166667;AN=60;AT=>8380>8381>8383>8380>8382>8383;NS=37;LV=0;ID=1-176501-SNV->8380>8382>8383-1	GT
1	176504	>8383>8386	C	G	60	.	AC=5;AF=0.0833333;AN=60;AT=>8383>8384>8386>8383>8385>8386;NS=37;LV=0;ID=1-176504-SNV->8383>8385>8386-1	GT
1	176506	>8386>8398	GTC AAG	GCTGTG,GCTGCC ,TTCGC,TTACG	60	.	AC=1,2,44,1;AF=0.0166667,0.0333333,0.733333,0.0166667;AN=60;AT=>8386>8388>8389>8390>8391>8397>8398>8386>8388>8392>8393>8396>8397>8398>8386>8388>8392>8393>8394>8395>8398>8386>8387>8389>8393>8394>8397>8398>8386>8387>8389>8390>8394>8397>8398;NS=37;LV=0;ID=1-176507-COMPLEX->8388>8392>8393>8396>8397-4,1-176507-COMPLEX->8388>8392>8393>8394>8395>8398-5,1-176506-SNV->8386>8387>8389-1,1-176509-COMPLEX->8389>8393>8394>8397-2,1-176506-SNV->8386>8387>8389-1,1-176510-SNV->8390>8394>8397-1	GT
1	176513	>8398>8401	G	A	60	.	AC=1;AF=0.0166667;AN=60;AT=>8398>8399>8401>8398>8400>8401;NS=37;LV=0;ID=1-176513-SNV->8398>8400>8401-1	GT
1	176516	>8401>8406	TA	TC,CA	60	.	AC=3,45;AF=0.05,0.75;AN=60;AT=>8401>8403>8404>8406>8401>8403>8405>8406>8401>8402>8404>8406;NS=37;LV=0;ID=1-176517-SNV->8403>8405>8406-1,1-176516-SNV->8401>8402>8404-1	GT
1	176519	>8406>8410	C	T,A	60	.	AC=3,45;AF=0.05,0.75;AN=60;AT=>8406>8408>8410>8406>8409>8410>8406>8407>8410;NS=37;LV=0;ID=1-176519-SNV->8406>8409>8410-1,1-176519-SNV->8406>8407>8410-1	GT

An example of bubbles categorized into variations

```
1 176506 >8386>8398 GTC AAG >8386>8388>8389>8390>8391>8397>8398
GCTGTG 1-176507-COMPLEX->8388>8392>8393>8396>8397-4
GCTGCC 1-176507-COMPLEX->8388>8392>8393>8394>8395>8398-5
TTCGCG 1-176506-SNV->8386>8387>8389-1;1-176509-COMPLEX->8389>8393>8394>8397-2
TTCACG 1-176506-SNV->8386>8387>8389-1;1-176510-SNV->8390>8394>8397-1
```

**Q25.** 15. Page 4 line 113: It is odd that there are so many more insertions than deletions. I'm
curious if the authors have any theories.
In our study, insertion counts were similar to deletion counts, but by total length, insertions
greatly exceeded deletions, consistent with recent human pangenome analyses (Schloissnig et al.
2024). Similarly, a recent cattle pangenome analysis incorporated 654,317 nonredundant SVs
into the final graph, including 187.23 Mb of genomic deletions and 265.16 Mb of non-reference
sequences (≥ 50 bp), again showing longer total insertion sequence than deletions (Zhao et al.,
GBP, accepted). This trend likely reflects reference bias from using the Hereford ARS-UCD 2.0
genome (which lacks many Holstein-specific sequences classified as insertions), the presence of
multiple distinct alleles for insertions versus the single "absent" allele for deletions, higher
sensitivity of graph-based frameworks—especially when paired with long-read sequencing—for
detecting novel sequences, and biological drivers such as transposable element activity and
lineage-specific expansions. After excluding ARS-UCD 2.0 sequences, our H20D pangenome
graph contained 51.01 million non-reference nodes and 69.14 million edges. We acknowledge
that some orthologous insertions may be evolutionarily linked, and further work—including
additional long-read datasets and validation with orthogonal technologies—will be needed to
assess their novelty and clarify their origins.

Q26. 16. Page 9 line 335: How were the ancestral alleles determined?

AU: In our analysis, ancestral alleles were inferred based on the three major allele types identified across the 40 high-quality assemblies. For each variant, the allele shared by all three groups was designated as the putative ancestral allele. While this approach does not incorporate an outgroup species, it serves as a practical proxy by leveraging shared variation patterns among closely related individuals. On L328, we added “the allele shared across all three groups was designated as the putative ancestral allele”.

Q27. 17. Page 9 line 353: It was unclear what was being compared here- based on the figure, it seems to be the percent of variants called on H20D using their MC method that were also called by PanGenie, and vice versa?

AU: Figure 5a compares variant call concordance between two methods on the H20D dataset. The middle percentages on one axis show the proportion of variants called by our MC method (83.54%) that were also detected by PanGenie (69.79%). The top (16.46%) and bottom (30.21%) percentages represent variants uniquely called by the MC method or PanGenie, respectively.

Q28. 18. Page 9 line 353: A better pangenome reference improves SV genotyping accuracy as compared to what? Only one pangenome reference was used.

AU: The statement refers to improvements in SV genotyping accuracy achieved by using a more comprehensive pangenome reference compared to read-based SV callers using a linear reference genome (Figure 5d). While only one pangenome reference was used in this study, it represents an improvement over traditional single-reference approaches by incorporating greater genetic diversity and structural complexity (Figure 5d). On L347, we modified to “underscoring the value of a pangenome reference in improving SV genotyping accuracy over a standard linear reference.”

Q29. 19. Page 10 line 382: When comparing PanGenie on the J10D graph with the calls from H20D, the calls in common were reported as an overall count; could this be reported as a percent as well, as it is for the other methods?

AU: We did not aim to directly compare PanGenie results for the 20 Holsteins using different pangenome references (J10D vs. H20D) because such a comparison would be confounded by differences in both sample size and the structure of the Holstein-specific pangenomes. Instead, we used nine alternative strategies as background references to provide a fairer assessment of consistency rates. This has been clarified in the revised manuscript on L373, “We did not directly compare PanGenie results for the 20 Holsteins using H20D vs. J10D, as differences in sample size and pangenome structure would confound results. Instead, nine alternative strategies were used as background references to provide a fairer assessment of consistency rates.”

Q30. 20. Page 12 line 466: The SNP-based GWAS was not described.

AU: We appreciate the reviewer’s comment. In fact, the SNP-based GWAS was conducted using the same methodology as the SV-based GWAS. To clarify this, we will update the Methods section by changing the heading from “Genome-wide association study between SV and phenotypes” to “Genome-wide association study between SV/SNP and phenotypes,” and revise the text accordingly to explicitly describe the analysis for both variant types.

**Q31.** 21. Page 13 line 483: I'm not sure if the SVs can be classified as "breed-specific" if they
were not checked for in other breeds.

**AU:** We changed "breed-specific" to "within-breed" to better convey our intended meaning.

**Q32.** 22. Page 3 line 70: "we constructed and compared breed-specific (H20D) with multiple
other... -> "a breed-specific pangenome (H20D)"

**AU:** Corrected.

**Q33.** 23. Page 7 line 239: "An average of 16.24k SVs was identified" -> "were identified"

**AU:** Corrected.

**Q34.** 24. Page 8: Many references to Figure 4 are missing or incorrectly reference Figures 3 or
S9.

**AU:** Good catch! We have fixed these errors. See our answer to Q10.

**Q35.** 25. Page 8 line 296/Figure 4a: The text says that Sniffles detected two deletions but in the
figure there appears to only be one. Similarly, the insertion seems to be two insertions in the
graph.

**AU:** The region in question contains tandem repeats of TGTGTG or GTGTGT motifs, which
complicates read mapping and SV detection due to repetitive sequence and mapping score
differences, as illustrated in Figure 4b and Figure S10. Sniffles detected two closely located
deletions: one of 391 bp starting at position 1,579,558 and another of 422 bp starting at
1,579,527. However, due to differences in read alignment scoring by Minimap2, these calls
likely represent the same underlying deletion event. In Figure 4a, this is shown as a single
deletion to avoid confusion, but both deletion calls are acknowledged in the text and Figure 4b to
explain the mapping ambiguity. Similarly, what appears as two insertions in the graph reflects
alternative representations of a complex insertion event within the tandem repeat region (Figure
4b: 32 bp, 132bp, and 166 bp). On L314-318, we clarified this in the manuscript to improve
consistency between the figure and text.

**Q36.** 26. Figure 4c: I found the variants floating in the middle of the graph to be confusing.

Could the paths on the graph be highlighted a different way? Maybe an image from the sequence
tube map could be more helpful?

**AU:** The sequence tube map in our original Figure 4c was indeed quite large and visually
cluttered. We have revised the figure by moving the variants outside the circle, which improves
clarity and readability.

**Q37.** 27. Figure 4d: I don't think the orientation of the nodes matters, since there don't appear to
be any inversions. The "30bp INS" and "32bp INS" arrows also seem to be switched.

**AU:** The orientation of the nodes is important because the sequences represented by these nodes
are reverse complements in the graph, not simply forward strands. This reversal affects how
insertions and their precise locations are interpreted. Regarding the "30 bp INS" and "32 bp INS"
arrows, they were mislabeled and we switched their labels.

**Q38.** 28. Page 8 line 304: "Unlike other pangenomes" but only the haploid pangenome was
compared.

AU: We deleted “, unlike other pangenomes.”

**Q39.** 29. Page 9 line 310/Figure 4c: The path of the insertion in the text doesn't match the path
in the graph

AU: “the INS and DEL variants corresponded to nodes starting at 1,072,043 and ending at
1,072,086 (Figure 4c). The INS followed the path >1072056>1072057>1072058”. The “nodes
starting at 1,072,043 and ending at 1,072,086” refer to the large path for the variation location,
which includes other variations such as 30/32 bp INS and SNVs. Note that we shortened the
coordinates by removing “10720”, leaving only the last two digits, like 43 to 86.

**Q40.** 30. Page 10 line 396: PG(JerRef) is not named consistently with the other methods

AU: We appreciate the reviewer's attention to detail, including the use of this nomenclature. The
label PG(JerRef) was intentionally defined as a special case to distinguish it from the other name.
Specifically, while PG(WGS) denotes imputation using the Holstein-based pangenome with 20
Holstein WGS samples as reference, PG(JerRef) refers to the same framework but with the
Jersey pangenome used as the SV reference panel instead of the Holstein pangenome. Thus, the
distinct naming highlights this substitution of the reference SV panel, while retaining the same
WGS dataset.

**Q41.** 31. Page 10: I think this is missing/mixing up references to Figure 6.

AU: Figure 6 was mislabeled as Figure S15. We corrected them.

**Q42.** 32. Page 11 line 431: “multiple SV” -> “multiple SVs”

AU: Corrected.

**Q43.** 33. Discussion: The first sentence of the discussion is repeated.

AU: Deleted.

**Q44.** 34. Figure 1: I was confused by panels a and c. They appear to be showing the same data,
but with different results: the shared length goes down to ~2.7G in panel a and ~1.8G in panel c.

AU: Panel 1a shows sample-level calculations, where the two haplotypes from each individual
are combined before measuring shared length. Panel 1c shows assembly-level calculations,
where each haplotype assembly is analyzed separately. This difference in calculation explains
why the total shared length is ~2.7 Gb in panel a but ~1.8 Gb in panel c. We have clarified this
distinction in both the figure legend and the main text.

**Q45.** 35. Figure 2: Panel a is missing an x-axis label

AU: We added “SV count” to the x-axis label to represent the event count in thousands, with
zero positioned at the center.

**Q46.** 36. Missing citations for Clair3 and DeepVariant

AU: They are added on L705.

**Q47.** 37. Page 17 line 657: The samples from CDDR were never mentioned in the paper

AU: On L416 and L726, we added two citations, which reported these 173 CDDR Holstein
samples.

Reviewer #4 (Remarks to the Author):

In this study, Yang et al. contribute to the growing body of work advocating for the use of pangenomes to genotype complex structural variants (SVs), which remain relatively understudied. The manuscript appropriately emphasizes the need to understand the genetic sourcing behind pangenome builds to effectively assess complex SVs. While the authors consistently apply previously published methods, the frequent switching between phased, unphased, and mixed-breed pangenome builds complicates interpretation and makes it difficult to extract clear conclusions. The extensive number of analyses reflected in 13 and 30 supplemental figures and tables, respectively, that demonstrates a commendable thoroughness in method optimization. However, this also becomes a liability, as many comparisons yield overlapping results (e.g., more diverse pangenomes identify more SVs overall but may struggle with complex SVs), which obscures the key takeaways.

Q48. The manuscript would benefit from simplification, particularly through the removal or relegation of non-essential analyses. For example, the sections on SNV calling using pangenomes are less impactful, given that this is not a major advantage of pangenome references compared to linear T2T-like genomes. A concise summary of the SNV results in the main text, with detailed results moved to the supplement, would improve focus. Additionally, some of the conclusions on pangenome builds and SV genotyping reiterate findings already reported, such as those in Pausch et al., Genome Research (2024), which limits the novelty of the work. That said, this study does offer valuable insights, particularly regarding the differential performance of SV genotyping using high- versus low-diversity pangenome graphs.

AU: We have condensed the sections on SNV calling using pangenomes by moving detailed analyses and results to the Supplementary Materials, retaining only a concise summary in the main text. Regarding conclusions overlapping with prior work (e.g., Pausch et al., Genome Research 2024), we have carefully revised the manuscript to better highlight our novel contributions, particularly the insights into differential SV genotyping performance when using high- versus low-diversity pangenome graphs.

Other minor points by section:

Q49. Introduction

The introduction seems to lack a clear narrative to effectively set up the main theme of optimizing pangenome graph construction. I recommend a partial rewrite that more concisely frames the study within the context of prior work. This should include a summary of relevant studies in cattle particularly those from Pausch et al. as well as the key human pangenome efforts that define the current state of the art. Emphasis should be placed on short-read genotyping performance using pangenomes, drawing from foundational work by Garrison et al. and Paten et al. This would provide a stronger conceptual foundation and better contextualize the goals and significance of the current study.

AU: Thank you for this valuable suggestion. We agree that a clearer, more focused narrative in the Introduction will better set up the study's main theme of optimizing pangenome graph construction. We have revised the Introduction to concisely frame the study within the context of prior relevant work, including key cattle studies such as Pausch et al., and major human pangenome efforts that represent the current state of the art. Additionally, we emphasized short-read genotyping performance using pangenomes, referencing foundational studies by Garrison et

al. and Paten et al., to provide a stronger conceptual foundation. Also, related sentences in the
Discussion have been moved here.

**Q50.** p. 3 line 56- SVs have been shown to (need citations here).

**AU:** We added citations.

Results

**Q51.** p. 4. Line 109. I think this is the first use of the term COMPLEX as a classification for
some SV types. This needs to be defined at some point.

**AU:** See our answer to Q24 above.

**Q52.** p. 5 line 163-64. Briefly explain why SVs on ChrX from female samples were retained for
this analysis.

**AU:** We retained ChrX SVs from females to allow consistent comparison between the Holstein
pangenome (H20D) and the multi-breed pangenome (M13H), which includes both sexes and
varies in sex composition.

**Q53.** p. 6. Lines 87-191. After this conclusion of haploid limitations don't the authors focus on
diploid modelling for the remainder of the study.

**AU:** Yes, following the discussion of haploid model limitations, the study shifts to focus
primarily on diploid modeling for all subsequent analyses. We have clarified this transition in the
manuscript to ensure readers understand the rationale and the change in approach. On L244, we
added "Building on these insights, we next explore the advantages of diploid pangenome
modeling and its impact on SV detection and genotyping accuracy."

**Q54.** p. 7. Lines 225-26. This section can be condensed by deemphasizing long read mapping
consistency as long-read mapping inferiority is already established earlier.

**AU:** See our answer to Q12.

**Q55.** p. 9. Lines 332-345. This section can be moved to supplement and is an example of my
recommendation to move all the SNV characterizations to supplement.

**AU:** We apologize for mislabeling Figure 4 as Figure S9. This section primarily addresses SVs,
with some SNV content. To maintain focus on SV genotyping, we have kept it in the main text
while moving the details to Supplemental Notes.

**Q56.** p. 9. Line 346. This section is the most important narrative to expand in this reviewers
opinion even at the expense of other sections. Cohorts with vast numbers of short-read samples
are available for genotyping and GWAS study. The limitation in their use for SV assessment
needs to be better understood across species with different evolved structures, especially those
domesticated.

**AU:** We appreciate the reviewer's insight regarding the importance of this section. While expanding
the discussion on large cohorts of short-read samples for SV genotyping and GWAS would
indeed strengthen the manuscript, it falls beyond the scope of this study, as we have submitted a
companion manuscript on this topic, currently under review at Nature Communications
(NCOMMS-25-57451-T). Instead, on L536 in Discussion, we added "Expanding pangenomic

analyses across additional breeds and species—including dairy and beef cattle, swine, and
poultry—will not only broaden our understanding of SVs in domesticated animals but also
address key challenges of SV assessment across diverse genome architectures and evolutionary
histories, guiding future studies in diverse populations.”

**Q57.** p. 10. Line 390. Same argument as before for the movement of these analyses to the
supplement.

**AU:** This section focuses on SNV discovery. Since we have submitted a companion manuscript
to *Nature Communications* covering the SV and SNV imputation panel, we believe this section
will be important for future references. Accordingly, we have trimmed and retained it in the main
text, while moving additional details to the Supplementary Notes.

**Q58.** p. 11. Line 421. Use SV abbreviation here and note which genotyping method was used. I
assume PanGenie. Why were these 173 Holstein’s chosen?

**AU:** On L415, we added “To investigate the role of SVs in complex cattle traits, we used
PanGeine to genotype H20D pangenome-derived SVs in 173 Holstein cattle, which have been
reported before (Bickhart et al. 2016; Boschiero et al. 2024).”

**Q59.** p. 11. Lines 436-37. How so on their influence? Detection alone or similar agreements for
SNVs in other GWAS studies.

**AU:** The influence mentioned refers not only to the detection of SVs but also to their
demonstrated associations with phenotypes, supported by concordance with SNV-based signals
reported in previous GWAS studies. On L430, we modified to “These findings highlight that
SVs not only are readily detectable but also often co-locate with GWAS signals from SNVs,
indicating their potential functional impact on economically important traits in Holstein cattle.”

**Q60.** p. 11. Line 449. What does SV-related genes mean? Some clarification would be better.

**AU:** On L444, we added “We define SV-related genes as those that overlap or lie near SVs and
may be affected in dosage, regulation, or structure.”

**Q61.** p. 12. Lines 463-65. This statement in the results could perhaps be moved to discussion.
Same comment for lines 471-75.

**AU:** We have moved these passages to Discussion.

Discussion.

**Q62.** The discussion revisits themes explored in earlier studies on the use of pangenomes for
genotyping, with the primary novelty here being the specific application to cattle genomes.
However, the discussion lacks citations that contrast these findings with prior work, which limits
the reader's ability to contextualize the results. It is unclear why the authors chose not to
reference earlier studies, and doing so would strengthen the discussion by clarifying how this
work builds upon or diverges from established findings.

**AU:** We acknowledge that incorporating additional citations contrasting our work with prior
studies would strengthen the manuscript and provide better context for readers. Accordingly, we
have revised the Discussion by moving this section to the Introduction. In the updated
Introduction, we now reference key prior work on pangenome-based genotyping, highlighting

both similarities and distinctions, particularly in cattle genomes. This revision clarifies how our
study builds upon and diverges from existing literature, thereby emphasizing its contribution to
the field.

**Q63.** p. 13. I don't think the term practical applies here. It is currently not practical to genotype
thousands of Holsteins that have been sequenced at lower coverage, to address cost concerns, for
cost and imputed for SNVs.

**AU:** We revised "practical applications" to "future potential" to avoid overstating the current
feasibility of genotyping thousands of Holsteins sequenced at lower coverage, especially
considering cost constraints and reliance on imputation for SNVs.

**Q64.** p. 13. Lines 514-15. I think a statement that includes "offers improved accuracy" over
existing genome selection indices is premature until further studies prove this hypothesis. It
certainly could once models incorporate SVs for larger numbers of cattle.

**AU:** We deleted and replaced it with "Detecting within-breed and crossbreed SVs provides a
foundation for potentially improving genomic selection, as it may help breeders identify alleles
associated with economically valuable traits."

Methods.

**Q65.** p. 15. Line 553. Gao et al. under review needs clarification if this study will overlap
significantly.

**AU:** Gao et al. was published at <https://pubmed.ncbi.nlm.nih.gov/40258473/>. There is no
significant overlap with this manuscript.

**Q66.** p. 15. Line 562. Cite the study that refers to previously analyzed cattle genomes.

**AU:** Gao et al. 2025 was added.

**Q67.** p. 18. Line 717. What is this reference?

**AU:** Sorry! It was added.

**Q68.** p. 19. Lines 750-51. This sentence is confusing. Clarify the methods.

**AU:** On L749, we changed to "The significance test was performed using randomly selected SVs
and SNPs in equal numbers for the fat GWAS analysis."

**Q69.** I commend the authors for generating high-quality data using appropriate techniques,
which have been carefully analyzed, interpreted, and presented in detail. I did not identify any
major methodological flaws. However, the study would benefit from clearer messaging and more
explicit acknowledgments that many of the findings align with and confirm earlier work on
pangenome genotyping in other species. The primary significance of this work lies in providing
additional validation that can inform the computational design of future studies particularly those
aiming to enhance the efficiency of SV genotyping across large whole-genome sequencing
cohorts.

**AU:** We sincerely appreciate the reviewer's positive evaluation of our study's data quality,
methodology, and analysis. We acknowledge the suggestion to strengthen the manuscript's
messaging by explicitly recognizing that many findings align with and confirm previous work on

pangenome genotyping in other species. See our answers to Q49 and Q62. On L542, we also
added “It provides validation to guide the design of future studies, particularly those aimed at
improving the efficiency of SV genotyping in large whole-genome sequencing cohorts.”

Bickhart DM, Hutchison JL, Null DJ, VanRaden PM, Cole JB. 2016. Reducing animal
sequencing redundancy by preferentially selecting animals with low-frequency
haplotypes. *J Dairy Sci* **99**: 5526-5534.

Boschiero C, Neupane M, Yang L, Schroeder SG, Tuo W, Ma L, Baldwin RL, Van Tassell CP,
Liu GE. 2024. A pilot detection and associate study of gene presence-absence variation
in Holstein cattle. *Animals* **14**: 1921.

REVIEWERS' COMMENTS

Reviewer #2 (Remarks to the Author):

The authors addressed all my comments. I appreciate the GitHub repository, but it still needs some tweaking. For example, the "annotate_vcf.py" file is missing.

AU: Thank you for checking the repository carefully. We would like to clarify that our workflow does use the script `annotate_vcf.py`, but this script is authored and maintained by the PanGenie software team, not by us. Because the script is part of the official PanGenie pipeline, and due to authorship and licensing considerations, we did not copy or redistribute it in our own GitHub repository. Instead, as stated in the Methods section, we provide the official link to the script from the PanGenie developers' repository: https://github.com/eblerjana/genotyping-pipelines/tree/main/prepare-vcf-MC/workflow/scripts/annotate_vcf.py. All scripts developed in this study are included in our GitHub repository. The PanGenie script above is the only external component required, and our workflow runs correctly using the version provided by the PanGenie team.

Reviewer #3 (Remarks to the Author):

I thank the authors for the significant changes that they have made to the paper and for their thorough responses to my comments. I find this draft to be much more clearly written and the results presented in a more honest and impactful way. I have only a few remaining concerns.

1. Response to Q21: The majority of this paper is spent comparing methods, and resource consumption is an important consideration when choosing a method. I would still like to see an analysis of runtime and memory, even if it is only a rough estimate of only the MC pipeline mentioned briefly.

AU: While we did not record detailed per-step runtime or memory logs during the original analyses, we agree that computational requirements are important for evaluating and comparing methods. Although re-running the complete pipeline is not feasible at this stage, we now provide a rough estimate based on our actual computational environment. On L619, we added "For example, to construct H20D, we ran the MC pipeline using 64 CPU cores, with its sub-tasks configured to use 32 cores and approximately 8 GB of RAM per core. Under this configuration, the entire MC workflow required about one week to complete."

2. Page 10 line 395: In their response to Q16, the authors say that they did not designate a truth set. While they did show pairwise analyses in the figure and did not specifically designate a truth set, in the text they present F1, precision, and sensitivity values relative to the HB method, which I find to be implicitly making a claim about correctness of other methods based on the HB results. Although this is not an incorrect result, I find it to be very misleading unless it is clearly and explicitly described why it was done.

AU: We agree that using the HB method as a reference for reporting F1, precision, and sensitivity could be misinterpreted as implying correctness. Our intention was only to provide a

relative comparison, not to designate HB as the truth set. We have revised the text to explicitly clarify this point and to avoid any implication that HB represents a gold-standard method. On L412, we added “although HB results do not represent the gold-standard truth sets.”

3. Response to Q13. I believe the authors misinterpreted the analysis I suggested; the analyses they referenced in their response still perform pairwise comparisons between methods. I was suggesting pooling all variants from all methods and stratifying them by how many tools support each variant. Then compare each method of interest to the stratified set of pooled variants. However, while I still believe that this analysis would be relatively easy and informative, the authors have already mostly addressed my main concern that their analyses didn’t support their claims by softening their claims so this is less important.

AU: We appreciate the suggestion to pool all variants and stratify them by the number of supporting methods. We now understand that this differs from the pairwise comparisons we performed. While we agree that such an analysis would be informative, the main concern has been addressed through the revisions that soften our original claims. We therefore have not added this additional analysis but have clarified our reasoning in the revision.

4. Response to Q15. I agree with the authors that the variants from the H20D graph are a reasonable truth set here, but this should be explained explicitly in the text.

AU: We agree that the rationale for using the H20D graph variants as a truth set should be stated explicitly. On L221, we added “To highlight the distinctive properties and high-confidence variant representation of the pangenome approach, we used the H20D pangenome variants as a baseline truth set for pairwise comparisons.” Our intention here is not to compare the external methods to each other, but rather to contrast pangenome-based strategies with assembly- or mapping-based approaches.

5. Page 3 line 75: “Besides enhancing the efficiency and accuracy of genome assembly, variation-aware graphs...” this needs a citation. I am not aware of any such work.

AU: “Besides enhancing the efficiency and accuracy of genome assembly,” was removed.

6. Page 3 line 80: I would consider the graph itself to be the catalog. Also Giraffe is a mapper, vg call is the genotyper that is sometimes used with Giraffe.

AU: We agree that the graph itself should be described as the catalog, and that Giraffe is a mapper while vg call performs genotyping. We have updated the wording in the manuscript to accurately reflect this distinction on L79 “Tools like PanGenie⁴² and Giraffe (from vg)⁴³ were designed to map or genotype genetic variants, including SVs, using short reads and long-read-based SV catalogs⁴⁴”

7. Page 5 line 150: This could reference the methods for the definition of COMPLEX variants, and also for how multi-allelic variants are converted to bi-allelic.

AU: We have updated the text to reference the Methods section where the definitions of COMPLEX variants and the procedures for converting multi-allelic variants to bi-allelic form are described. On L154, we added ““COMPLEX” refers to variant alleles located within pangenome-graph bubbles containing more than two branches, representing sequence changes of at least two

base pairs that cannot be classified simply as DEL or INS'. On L160, we modified to "To account for nested and structurally complex variations, we used PanGenie's graph-based genotyping framework to decompose multi-allelic sites into standardized bi-allelic representations."

8. Page 6 line 209: There is an extra parentheses after "haploid assemblies"

AU: Corrected.

9. Page 9 line 317: Should this figure reference be 4a?

AU: Corrected.

10. Page 9 line 346: Is H20D referring to the MC pipeline with H20D? Additionally, these are both pangenome methods so this does not yet "underscore the value of a pangenome reference".

AU: We have clarified in the text that H20D refers to the MC pipeline using the H20D reference. We also slightly revised the wording to state that, although both methods compared here are pangenome-based, their performance relative to a linear reference does help demonstrate the broader value of using a pangenome reference.

11. Figures 2d, 2f, 6d, and 6f: What do the sizes of the squares/circles mean?

AU: The sizes and colors of the squares/circles reflect the same underlying values shown in the corresponding color bars. These are the default settings of the plotting program: larger, warmer-colored squares/circles represent higher values, while smaller, cooler-colored squares/circles represent lower values. We have added a brief clarification in the figure captions.

Reviewer #4 (Remarks to the Author):

In reviewing the authors' responses and how they addressed prior critiques, I find that the manuscript now presents a much-improved interpretation of pangenome applications and relevance. Some sections could still benefit from more concise writing, and a few results remain somewhat overstated in their novelty. Nonetheless, the manuscript provides a solid and informative account of how pangenomes can be effectively utilized.

AU: In reply to Author_Guidance_NCOMMS-25-25853A_1761856297_58, we have carefully addressed these critiques and moderated a few results that were previously slightly overstated in terms of novelty.